# A Field Evidence Model: How to Predict Transport in Heterogeneous Aquifers at Low Investigation Level?

Alraune Zech[1][2], Peter Dietrich[1][3], Sabine Attinger[1][4], and Georg Teutsch[1]

[1]Helmholtz-Centre for Environmental Research - UFZ, Leipzig, Germany
[2]Utrecht University, Department of Earth Science, Utrecht, The Netherlands
[3]Eberhard Karls University Tübingen, Germany
[4]University of Potsdam, Germany

**Correspondence:** Alraune Zech (a.zech@uu.nl)

**Abstract.** Aquifer heterogeneity in combination with data scarcity is a major challenge for reliable solute transport prediction. Velocity fluctuations cause non-regular plume shapes with potentially long tailing and/or fast travelling mass fractions. High monitoring cost and a shortage of simple concepts have limited the incorporation of heterogeneity to many field transport models up to now.

We present an easy-applicable hierarchical conceptualization strategy for hydraulic conductivity to integrate aquifer heterogeneity into quantitative flow and transport modelling. The modular approach combines large-scale deterministic structures with random sub-structures. Depending on the modelling aim, the required structural complexity can be adapted. The same holds for the amount of monitoring data. The conductivity model is constructed step-wise following field evidence from observations; seeking a balance between model complexity and available field data. Starting point are deterministic blocks, derived

from head profiles and pumping tests. Then, sub-scale heterogeneity in form of random binary inclusions is introduced to each block. Structural parameters can be determined e.g. from flowmeter measurements or hydraulic profiling.

    As proof of concept, we implemented a predictive transport model for the heterogeneous MADE site. The proposed hierarchical aquifer structure reproduces the plume development of the MADE-1 transport experiment without calibration. Thus, classical ADE models are able to describe highly skewed tracer plumes by incorporating deterministic contrasts and effects of

connectivity in a stochastic way without using uni-modal heterogeneity models with high variances. The reliance of the conceptual model on few observations makes it appealing for a goal-oriented site specific transport analysis of less well investigated heterogeneous sites.

## 1   Introduction

Groundwater is extensively used worldwide as the major drinking water resource and consequently needs to be protected with respect to quantity and quality. Increasing pressure on the quality originates from the intensification of agriculture using

agrochemicals (non-point sources), an increased urbanization with the resulting solid and liquid wastes and contaminant spills from industrial applications (point sources).

Essential for groundwater protection is the quantitative analysis of the fate and transport of various contaminants in the groundwater body. This can be either for a provisional risk assessment or for the clean-up of an already existing groundwater contamination. Numerical models are common tools to quantify the flow and transport, where partial differential equations are solved using initial and boundary conditions.

For simplicity, we restrict ourselves to saturated flow and transport of a dissolved, non-reactive contaminant. The governing equation for its concentration $C(\boldsymbol{x},t)$ is the advection-dispersion equation (ADE) (Bear, 1972):

$$\frac{\partial C(\boldsymbol{x},t)}{\partial t} = -\boldsymbol{u}(\boldsymbol{x},t) \cdot \nabla C(\boldsymbol{x},t) + \nabla \left(\mathbf{D} \cdot \nabla C(\boldsymbol{x},t)\right) \tag{1}$$

given in space $\boldsymbol{x} = (x,y,z)$ and time $t$. $\mathbf{D}$ is the dispersion coefficient tensor and $\boldsymbol{u}(\boldsymbol{x},t)$ is the Darcy velocity vector. The latter is a function of the hydraulic gradient $J$ and the heterogeneous hydraulic conductivity $K(\boldsymbol{x})$ through Darcy's Law. A proper description of the velocity field $\boldsymbol{u}(\boldsymbol{x},t)$, thus aquifer heterogeneity, is crucial for predicting the concentration distribution $C(\boldsymbol{x},t)$.

The adequate parametrization of the heterogeneous conductivity $K(\boldsymbol{x})$ poses a significant challenge in practical model setup due to data scarcity. Numerous deterministic and stochastic approaches have been developed to incorporate the effects of spatial heterogeneity of conductivity on flow and transport, particularly in the context of stochastic subsurface hydrology (Koltermann and Gorelick, 1996). Representing conductivity by an effective uniform value is convenient for aquifers of low heterogeneity since it can be inferred from pumping tests with decent monitoring effort. But predicting transport in aquifers of significant variability fails when neglecting local effects of heterogeneity and preferential flow.

Stochastic methods allow resolving heterogeneity and thus capture the induced uncertainty in flow and transport predictions. However, the amount of observation data required is usually high, depending on the method's complexity. Common methods are (i) Kriging (Kitanidis, 2008). (ii) Gaussian random fields (Dagan, 1989; Gómez-Hernández and Gorelick, 1989; Zinn and Harvey, 2003), potentially combined with Kriging for conditioning to observations; (iii) indicator/hydrofacies models (Journel and Gómez-Hernández, 1993; Carle and Fogg, 1996; Fogg et al., 2000); or (iv) multi-point statistics/training images (Strebelle, 2002; Renard et al., 2011; Linde et al., 2015).

For many unconsolidated sediments, field observations showed that conductivity is approximately log-normal (Delhomme, 1979; Gelhar, 1993; Rubin, 2003); characterized by the geometric mean $K_G$ and the log-conductivity variance $\sigma_Y^2$. Variogram analysis provides structural parameters such as correlation length $\ell$ and anisotropy ratio $e$ based on spatially distributed observations, e.g. from flowmeter, permeameter or injection logging. Despite increased efficiency in exploration methods, data is not even sufficient for variogram analysis in most practical cases thus hampering the practical application of Kriging and Gaussian random fields. Alternatively, hydrofacies models use indicator geostatistics with transition probability to generate geological heterogeneity structures. Although conceptually different, the required amount of input data is similarly high. Multi-point statistical methods provide heterogeneity structures of high geological realism, when training images are available. Although

satellite data might provide areal training images, vertical structures rely on extensive literature on geology or outcrop studies. Both are hardly available at the scale representative for plume transport impeding the method's use at hydrogeological sites.

A recent debates series (Rajaram, 2016; Fiori et al., 2016; Fogg and Zhang, 2016; Cirpka and Valocchi, 2016; Sanchez-Vila and Fernàndez-Garcia, 2016) outlined the gap between the advanced research in stochastic subsurface hydrology and its application in the practice of groundwater flow and transport modeling. We see a significant reason in the lack of data for complex stochastic models. Thus, we advocate the use of hierarchical approaches, combining deterministic and stochastic hydraulic conductivity conceptualization. Hierarchical approaches are regularly used in reservoir modelling (Damsleth et al., 1992; Smith et al., 2001; Bryant and Flint, 2009), particularly for consolidated sediments. Aside from qualitative approaches for multi-scale heterogeneity representation e.g. Neton et al. (1994); Herweijer (1997), or Koltermann and Gorelick (1996) (and references therein), only few quantitative approaches were proposed, such as: generating sequences of facies assemblages using indicator geostatistics and transition probability at various scales (Weissmann and Fogg, 1999; Proce et al., 2004), or combining training images for large-scale facies realizations with variogram-based geostatistical methods for random intrafacies permeability (Huysmans and Dassargues, 2009). Both approaches show a high level of model complexity and required (hydro-)geological input data.

Here, we present a parsimonious hierarchical heterogeneity conceptualization which is easy to apply in quantitative models for predicting flow and solute transport. In a deterministic-stochastic framework we combine descriptive zonation with statistical methods, following the lines of Gómez-Hernández and Gorelick (1989). Goal is to optimize the aquifer structure setup given the simulation target constrained by the available field data. Thereby, we aim to provide tools making aquifer heterogeneity more accessible for practical applications, including hands-on software. The approach is based on the fact that subsurface heterogeneity can be generally classified into (a) larger scale dominant features which primarily determine the general flow direction together with the average groundwater flow velocity; and (b) smaller scale features which are responsible for the dispersion, respectively the spatial spreading of a solute.

We create a deliberate connection between the model parameterization requirements and the field characterization methods beyond a single monitoring method. Pumping tests, for example, are best suited to determine the spatially averaged transmissivity respectively hydraulic conductivity, even in a heterogeneous aquifer environment (Herweijer, 1996; Zech et al., 2016). Together with head gradients from piezometric level maps this yields good estimates of the mean groundwater flow velocities. High resolution, small-scale borehole logs of hydraulic conductivity (e.g. from flowmeter or direct push methods) provide information on conductivity variability and consequently the dispersion parameters needed. Here, we consider two stochastic methods representing spatial variability: Gaussian random fields which require distributed observation data for a variogram analysis and a simplistic binary structure which relies only on a few (e.g. 2-4) well-logs, but takes parametric uncertainty into account. The latter is developed as option for less investigated sites only requiring a decent amount of field data from standard monitoring methods for heterogeneous aquifer modelling.

We demonstrate the methodology for MADE, a heterogeneous, well investigated research site (e.g. Boggs et al. (1990); Zheng et al. (2011); Gómez-Hernández et al. (2017)). Following our adaptive approach, we use various amounts and types of hydraulic observation data for heterogeneity conceptualization to construct numerical transport models. Predictions are

independently evaluated against field tracer data from the MADE-1 experiment (Boggs et al., 1992). We do not reconstruct the actual conductivity structure at MADE, but predict tracer plume behavior following a Monte Carlo approach devoid of calibration. Model results show good agreement with observed plume data, also compared to other predictive transport models for MADE (e.g. Salamon et al. (2007); Fiori et al. (2013, 2017); Bianchi and Zheng (2016)). In this line, we provide an alternative approach for predictive transport modelling at a significantly heterogeneous site with a simple conceptualization and decent observation effort.

The course of the paper is the following: section 2 features the approach in light of different modeling aims. Section 3 is dedicated to the application of the methodology for the MADE aquifer. We close with a summary and conclusions in section 4.

## 2 Approach

Large scale hydraulic structures of hundreds or more meters determine the groundwater flow direction and magnitude in combination with groundwater catchment boundaries. Subsequently, they set the mean transport velocity. This is the key parameter to predict the location of the bulk mass of substances dissolved in the groundwater when input conditions are known.

Variations of hydraulic properties on intermediate scale, in the range of tens of meters, generate spatially variable flow fields. They also render transport velocities variable at these scales resulting in a larger spreading of plumes. This is particularly important for modeling tailing or leading mass fronts. Fluctuations on scales smaller than these intermediate scales have a blending effect, generally increasing local mixing and enhancing dispersion (Werth et al., 2006).

Following this conceptual view, we generate hydraulic conductivity fields composed of three components: Module (A), (B) and (C) which capture the effects at large, intermediate and small scale heterogeneity, respectively. Each component is selected according to the model aim and the data at hand to parametrize the hydraulic conductivity for this component.

The procedure is exemplified for the MADE site. This significantly heterogeneous site was intensively investigated with various measurement devices providing many different data sets, as pumping tests, flowmeter and DPIL measurements (Boggs et al., 1990; Bohling et al., 2016). Detailed information on MADE can be found in section 3 and the *Supporting Information*.

In the approach, we considers several steps:

1. Specifying the aim of the model: What do we want to predict?

2. Selecting processes and process components which need to be accounted for in the model: What does this imply for the conceptualization of hydraulic conductivity?

3. Selecting suitable measurement methods: Which method can deliver the data needed for parameterizing hydraulic conductivity with affordable effort?

4. Conceptualizing hydraulic conductivity.

5. Calculating flow and transport.

Before specifying the hydraulic conductivity component Modules (A), (B) and (C), we illustrate our concept discussing two exemplary model aims.

## 2.1 Exemplary Model Aims

**Model Aim "Mean Arrival"**

1. **Aim**: Prediction of mean arrival of a contaminant from a point source.

2. **Processes:** Estimation of regional groundwater movement, direction and magnitude of flow making use of the groundwater flow equation and Darcy's law. Transport is modelled by advection. For sake of simplicity we do not consider reactivity.

3. **Field characterization:** Regionalized groundwater level measurements provide direction and magnitude of hydraulic gradient. It is critical to outline areas of different gradients (zones) indicating regional hydraulic conductivity trends and large scale heterogeneity. Pumping tests can provide independent values of effective transmissivity within each zone.

4. **Conceptualization of hydraulic conductivity:** Conductivity is considered homogeneous within each large scale zone. Effects of heterogeneity are captured in effective parameters representing average flow behavior, e.g. determined from pumping tests.

5. **Solving flow and transport:** Flow is solved either analytically, e.g. for one or two zones of different effective hydraulic conductivity, or numerically in case of a more complex spatial distribution of zones. Transport can be determined making use of analytical or numerically solutions of the ADE according to initial and boundary conditions.

### 2.1.1 Example MADE

The piezometric surface map of MADE (Boggs et al., 1992, Fig. 3) shows a significant non-uniform hydraulic head pattern. At 20 m downstream of the injection location, head isolines reduce abruptly. The reproduced head contours in Figure 1a allow to delineate two major zones: an area of low conductivity upstream (left) and high conductivity downstream (right). Two large scale pumping tests confirm the contrast in mean conductivity of about two orders of magnitude (Boggs et al., 1992). Consequently, flow should be modelled with distinct mean conductivity in two vertical zones (Figure 1b) when aiming to model mean arrival times for the MADE site.

**Model Aim "Risk Assessment"**

1. **Aim:** Prediction of early or late arrival of contaminants commonly used in risk assessments.

2. **Processes:** Flow and transport equations; it is particularly relevant to capture variability in transport velocity to estimate spreading behavior of plumes.

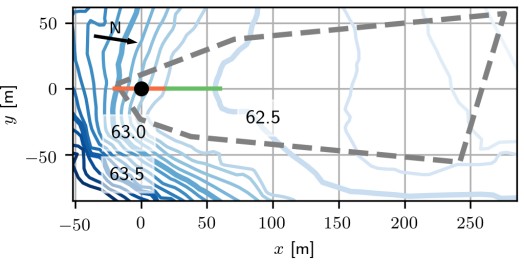
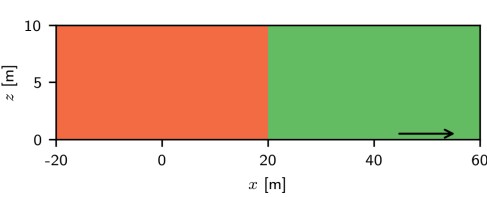

**Figure 1.** Left: Potentiometric surface map of head measurements according to Boggs et al. (1990). Orange-Green line indicates location of cross section displayed right: Concept (Module A) for large conductivity structure with deterministic zones of low (orange) and high (green) conductivity. Arrow indicates flow direction. Location of the interface between structures corresponds to change in hydraulic head pattern at $x = 20m$.

3. **Field characterization:** Detecting and delineating high and low conductivity subsurface structures with a characteristic horizontal length scale of several meters. Typical examples are channels formed in braided river systems. Typical investigation methods giving field evidence of such heterogeneity structures are small scale slug tests, borehole flowmeter logs or permeameter tests detecting strongly vertically varying conductivity.

4. **Conceptualization of hydraulic conductivity:** Spatially structured non-uniform conductivity.

5. **Solving flow and transport:** Small variations in conductivity allow to apply analytical solutions with effective measures, e.g. from first order theory (Dagan, 1989). Spatially resolved heterogeneity requires numerical solution of flow and transport with numerical tools (Monte Carlo approach).

#### 2.1.2   Example MADE

Borehole flowmeter logs at MADE (Rehfeldt et al., 1989; Boggs et al., 1990) reveal horizontal layers with conductivity differences over $2 - 3$ orders of magnitude. For instance, the flowmeter log *F-40* shown in Figure 2a has a bulk of high conductivity values with about $15\%$ of values being two orders of magnitude smaller. Logs at other locations (*F-09* and *F-18*) show the inverse behavior: a bulk of low conductivity values with embeddings of high conductivity.

Such strong vertical variation indicate the presence of high conductivity channels acting as preferential flow path and low conductivity zones with stagnant flow which both impact strongly on plume spreading behavior. Consequently, when aiming to model early and late plume arrival these feature need to be accounted for in a flow and transport model for the MADE site.

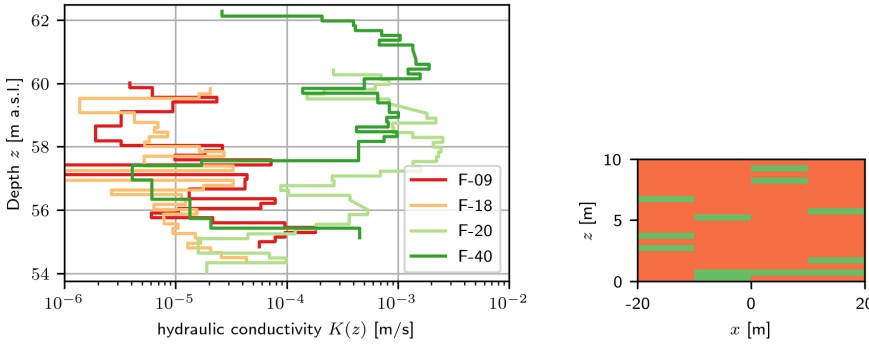

**Figure 2.** Left: Four flowmeter logs of hydraulic conductivity $K(z)$ versus depth $z$; the logs *F-09* and *F-18* are close to the tracer test injection location; *F-20* and *F-40* are several tenth of meters downstream (see Figure 3). Right: Concept of binary inclusion structure (Module B) with $15\%$ high conductivity inclusions (green) embedded in the bulk of low conductivity (orange). Inclusion length are arbitrarily chosen as $I_h = 5\,\mathrm{m}$ and $I_v = 0.5 - 1\,\mathrm{m}$.

## 2.2 Scale-dependent Conductivity Modules

Given the scale-dependency of hydraulic conductivity features and their distinct relevance for flow and transport predictions, we propose three components: Module (A), (B) and (C) which capture large, intermediate and small scale heterogeneity effects, respectively. Given a certain model aim, components are selected (or not) with regard to the available field data. We shortly discuss the Modules and motivate their use based on the data of the MADE site example for different aims.

**Module A**

The aquifer domain of interest is divided into deterministic zones of significantly different mean conductivity (i.e. more than one order of magnitude). The structure can comprise horizontal or vertical layering simply in blocks or complex zone geometries depending on information available. The use of Module A is warranted when observation data indicates significant areal conductivity contrasts.

The zones represent large scale geological structures exhibiting conductivity differences potentially over several orders of magnitude as a results of changes in deposition history or changes in the material's composition (Bear, 1972; Gelhar, 1993). Zones can be delineated using geologic maps, piezometric surface maps and geophysical methods providing information on aquifer structure, sedimentology and genesis. Pumping tests are suitable for identifying mean conductivities for each zone due to their large detection scale. Flow simulations on the deterministic zone structure should reproduce the observed head pattern.

The MADE site is an example where the concept of two zones of different mean hydraulic conductivity (Figure 1b) can reproduce conceptually the hydraulic head pattern. Details will be discussed in section 3.

**Module B**

When hydraulic conductivity shows heterogeneous features at the same length scale as the plume transport itself, they require proper resolution. A contaminant plume typically passes several of these intermediate scale features but not enough to ensure ergodic transport behavior. Thus, using effective parameters is not warranted. Since limited data availability precludes from a deterministic representation of these features, stochastic approaches suit best.

Binary stochastic models are a simple way to capture the effects of intermediate scale features (Haldorsen and Lake, 1984; Dagan, 1986; Rubin, 1995). Figure 2b shows an example how to conceptualize a medium with two $K$ values: inclusions ($K_2$) are embedded in the bulk conductivity ($K_1$), with $p$ characterizing the percentage of $K_2$. Inclusions of high conductivity may represent preferential flow paths whereas inclusions of low conductivity can be obstacles like clay lenses.

The inclusion topology is a matter of choice and data availability. A simple design is a distribution of non-overlapping blocks with horizontal length $I_h$ and thickness $I_v$. Figure 2b provides an impression with arbitrary choice of parameters. More complex layering structures can be adapted if additional topological information is available. However, the specific topology often plays a subordinate role. When not having any information on spatial correlation of heterogeneity, it is beneficial to assume some instead of sticking to a homogeneous model.

Characteristic length scales in vertical direction $I_v$ are detectable with low effort from a few borehole logs (Figure 2a). Characteristic horizontal length as $I_h$ are critical since they require spatially distributed observations. A parametric uncertainty approach can keep the effort low. A range of reasonable $I_h$ values is estimated and applied in the random inclusion model. A sensitivity analysis reveals the impact of the parametric uncertainty of $I_h$ on transport results. The estimates of $I_h$ could results from auxiliary data such as vertical length scale in combination with anisotropy ratios. Another option is expert knowledge based on geological structures and similarities to outcrop studies. Methods such as diffusivity tests (Somogyvari et al., 2016) or novel approaches for pumping test interpretation (Zech et al., 2016) also offer options to gain estimates for $I_h$.

The binary structure as in Figure 2b is beneficial in its plain stochastic concept relying on few input data, simple implementation and low computational requirement. It can be combined with Module (A) by implementing it within every deterministic zone preserving the mean conductivities. As for MADE, the inclusions represent the contrasting vertical layers as observed in flowmeter logs (Figure 2a), from which the inclusion parameters can be deduced for every deterministic zone (section 3).

**Module C**

Variations in grain size and soil texture form small scale heterogeneities of characteristic length scales up to one meter. Their relevance for transport predictions depends on the degree of heterogeneity and ergodicity. A plume is considered ergodic when the behaviour within one realization is statistically representative, i.e. exchangeable with ensemble behaviour. Figuratively speaking, an ergodic plume has travelled long enough to sufficiently sample heterogeneity. This is usually assumed for transport distances of $10 - 100$ characteristic lengths (Dagan, 1989), with higher values for increasing degree of heterogeneity. When ergodic, effective parameters can capture effects of heterogeneity. Otherwise, the use of a spatial random representation is warranted.

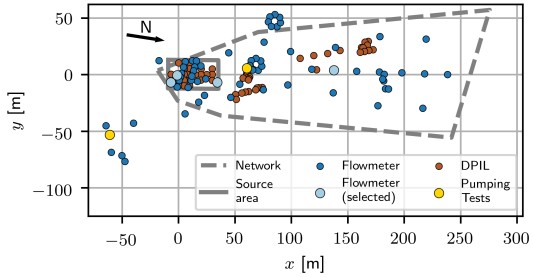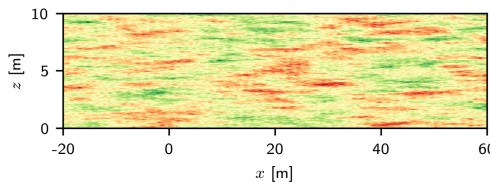

**Figure 3.** Left: Locations of measurements and tracer test observation network according to Boggs et al. (1990); Bohling et al. (2016). Right: Gaussian random field with exponential co-variance structure as conceptual module for small scale conductivity (Module C).

If required, small scale features can be conceptualized with a log-normal conductivity distribution $K(\boldsymbol{x}) \propto \mathcal{LN}(K_{\mathrm{G}}, \sigma_Y^2)$ with geometric mean $K_{\mathrm{G}}$ and log-variance $\sigma_Y^2$. Including a spatial correlation structure depends on the acquired complexity and the availability of two-point statistical data as correlation length and anisotropy. Figure 3b gives an example.

Geostatistical parameters can be inferred from spatially distributed observations (Figure 3a), e.g. permeameters, borehole flowmeter, or injection logging (Figure 4). This is related to high effort and costs. Novel techniques like DPIL (Dietrich et al., 2008; Bohling et al., 2016) can provide a large amount of data at acceptable costs and time, but they are only accessible for shallow sites. Alternative approaches derive geostatistical parameters directly from pumping tests (Zech and Attinger, 2016; Zech et al., 2016) or dipole tracer test (Zech et al., 2018). Note the discrepancy in geostatistical estimates among observation methods (Figure 4), which is not uncommon for heterogeneous sites. Differences are attributed to scale effects as a results of different method characteristics, such as support volume and resolution. This underlines the caution that has to be given to the appropriate use of observation data in conductivity conceptualization.

When combining with larger heterogeneity structures, small scale fluctuations are subordinate. In case of field evidence, Module (C) can be combined with Modules (A) and (B) by adding zero-mean fluctuations. According to Lu and Zhang (2002), the variances of heterogeneous sub-structures is additive. Thus, the log-normal variance relates to a 'variance gap' between the total variance, e.g. from a geostatistical analysis of the entire domain, and the binary model's variance (Module B). It can be interpreted as the system's variance which is not captured by intermediate and large scale heterogeneity. The length scales for a correlation structure should be significantly smaller than the inclusion lengths of Module (B). Including small-scale heterogeneity enhances the realism of conductivity structure – however, on the expanse of increasing investigation costs.

The MADE site is a rare example with geostatistics from multiple observation methods (Figures 3a and 4). Methods well suited for small scale heterogeneity show large variances from $4.5$ up to $5.9$. Given the high variance and the low mean conductivity, ergodic conditions cannot be assumed for transport within the range of a few hundred meters.

The large value in variance, as determined for MADE, can likely be the result of preferential flow and/or trends in mean conductivity. Thus, explicitly representing deterministic zones (Module A) and preferential flow paths (Module B) might render

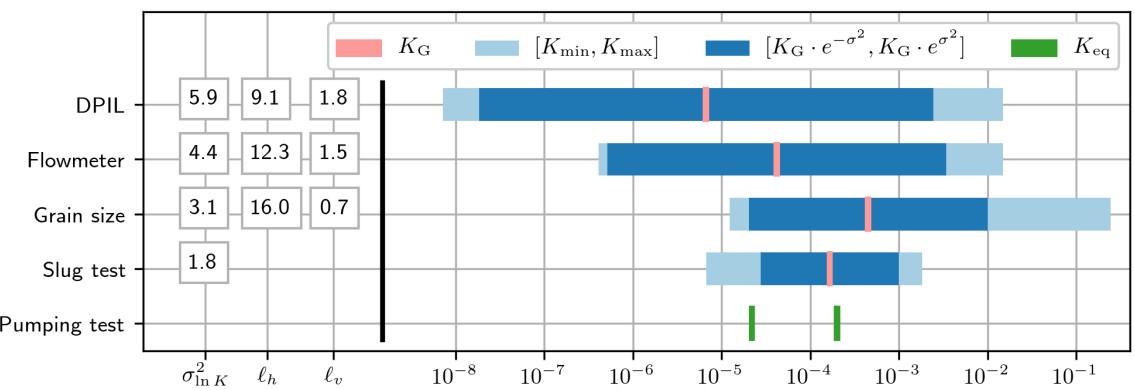

**Figure 4.** Geostatistical measures for MADE from DPIL (direct push injection logging) (Bohling et al., 2016), flowmeter, grain size analysis, slug tests (Rehfeldt et al., 1992) and effective mean values ($K_{eff}$) of two large scale pumping tests (Boggs et al., 1990): log-conductivity variance $\sigma^2_{\ln K}$, horizontal and vertical correlation length $\ell_h$ and $\ell_v$, respectively. Visualization of range of observed values from minimal ($K_{min}$) to maximal ($K_{max}$), variance range and geometric mean $K_G$.

the representation of small scale features (Module C) redundant. Modeling hydraulic conductivity as log-normal fields solely based on Module (C) seems warranted when there is no indication for deterministic zones or preferential pathways.

## Hierarchy of Scales

The hierarchy of scales poses an inherent problem for each groundwater model based on heterogeneous field data. Data interpretation often does not allow to clearly distinguish general trends from randomness. The three modules provide a simple classification of transport relevant heterogeneity scales: (A) beyond plume scale, i.e. above 100m; (B) range of plume scale (about 10-100m); and (C) sub-scale (<1m). This classification might not hold for every field and transport situation, but provides an orientation for developing site-specific heterogeneous conductivity structures.

Which module to integrate at a specific site depends on multiple aspects: (i) Is there field data evidence for a heterogeneity structure of a certain length scale?; (ii) Is there sufficient data to parameterize a conceptual heterogeneity representation? And (iii) is it necessary to present the heterogeneity given the travel distance of the plume (ergodicity)? Having a positive answer to each of the question for a certain module warrants its consideration in the conductivity conceptual model.

## 3 Predictive Transport Model for MADE

We validate our approach by performing flow and transport calculation for the MADE setting without parameter calibration. Although, many approaches to model the transport at the MADE site exist, including detailed aquifer conceptualizations (e.g.

Herweijer (1997); Julian et al. (2001), for a detailed review see (Zheng et al., 2011)), only few of them have a predictive character, i.e. devoid of calibration to transport results (Fiori et al., 2013, 2017; Dogan et al., 2014; Bianchi and Zheng, 2016).

Based on the scale-dependent conductivity modules (section 2.2), we develop different conductivity structures according to the field evidence given the structural data at MADE. We thereby aim to identify the "most simple" of our concepts which still provides a reasonable prediction of the complex observed mass distribution. The computed tracer plumes are compared to the MADE-1 transport experimental results (Boggs et al., 1992; Adams and Gelhar, 1992). Since the observed spatial concentration distribution is not available, we make use of 1D longitudinal mass transects at specified times.

Following the approach steps outlined in section 2, we define our model aim broader then specified in section 2.1: The target is predicting the general plume behavior. This might serve different purposes as e.g remediation and includes the mean flow behavior. The fact that there is no break-through curve data available for MADE, inhibits to study the subject of arrival times. Particularly critical is first arrival as discussed in Adams and Gelhar (1992). Processes involved here are flow and transport governed by Darcy's Law and the Advection-Dispersion-Equation (Eq. 1).

## 3.1 MADE Field Data

The MADE site is located on the Columbus Air Force Base in Mississippi, U.S.A. The aquifer was characterized as shallow, unconfined, of about $10 - 11\,\mathrm{m}$ thickness (Boggs et al., 1992). It consists of alluvial terrace deposits composed of poorly sorted to well-sorted sandy gravel and gravely sand with significant amounts of silt and clay. The first extensive field campaign by Boggs et al. (1990) yielded a multitude of hydro-geological information, as e.g. piezometric surface maps and hydraulic conductivity observations from soil samples, flowmeter and pumping tests (Figure 4). Field campaigns in subsequent years supplemented observations and data interpretations. For an overview see e.g. Zheng et al. (2011); Bohling et al. (2016) or Table 1 in the *Supporting Information*. We apply a porosity of $0.31$. Recharge is assumed uniform and very small (Boggs et al., 1990). Both quantities are kept constant due to the dominating effect of hydraulic conductivity given the significant variations and the uncertainty associated with observations (Figure 4).

The MADE-1 transport experiment was conducted in the years 1986–1988 (Boggs et al., 1990, 1992; Rehfeldt et al., 1992; Adams and Gelhar, 1992). A pulse of bromide was injected over a period of $48.5h$ applying a flow rate of $3.5\,\mathrm{l/min}$. The forced input conditions enlarged the tracer body at the source. Transport then took place under ambient flow conditions.

Concentrations were observed within a spatially dense monitoring network at several times after injection. We focus on the reported longitudinal mass distribution of Adams and Gelhar (1992, Fig.7) at six times: 49, 126, 202, 279, 370, and 503 days after injection. Values are integrated measures over transverse planes and accumulated over slices of $10\,\mathrm{m}$ length, given at the centers of slices at $-5\,\mathrm{m}$, $5\,\mathrm{m}$, $15\,\mathrm{m}$, .... The reported mass does not display mass recovery except at $126$ days with recovery rates of $2.06, 0.99, 0.68, 0.62, 0.54$, and $0.43$, for the six times, respectively. We do not normalize the reported mass to recovered mass, but stick to the actually observed values associating the mass loss to insufficient sampling in the downstream zone as discussed in details by Fiori (2014).

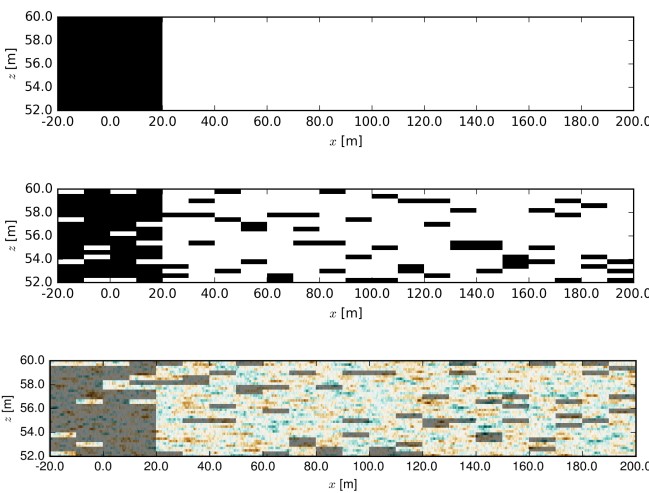

**Figure 5.** Realizations of hydraulic conductivity structures: (top) Deterministic zones (Module A), low $K_1$ in black, high $K_2$ in white. (center) Inclusions in deterministic zones (Modules A+B); amount of inclusions $p = 15\%$, inclusion lengths $I_h = 10\,\text{m}$, $I_v = 0.5\,\text{m}$. (bottom) Inclusions in deterministic zones and sub-scale heterogeneity (Modules A+B+C); correlation lengths $\lambda_h = 2.5\,\text{m}$, $\lambda_v = 0.125\,\text{m}$.

## 3.2 Hydraulic Conductivity Structures

Three hydraulic conductivity conceptualizations are designed in line with the specifications for MADE in section 2, which serve different model aims. Modules (A), (B) and (C) are combined successively to capture the scale hierarchy of heterogeneity at the MADE site. Figure 5 illustrated examples for each conceptualization.

### 3.2.1 Deterministic Zones (A)

Following the lines of Rehfeldt et al. (1992), we create conductivity zones based on the changes in the piezometric surface map (Figure 1). We chose two vertically arranged deterministic zones (Figure 5): a low in average conductivity zone $Z_1$ from upstream of the tracer input location to $x = 20\,\text{m}$ downstream and zone $Z_2$ as high-in-the-average conductivity area from 20 m downstream of the source (section 2.1.1).

We fix mean conductivity values in the zones as $\bar{K}_{Z1} = 2e - 6\,\text{m/s}$ and $\bar{K}_{Z2} = 2e - 4\,\text{m/s}$ with a contrast of two orders of magnitude as stated by Boggs et al. (1992). The specific values are chosen according to the two large scale pumping test (Boggs et al., 1992) and the head level rise during injection which is particularly important for early plume development. Details are given in the *Supporting Information*.

When fixing regional conductivities from pumping tests, model scale coincides with measurement scale. This way, our structures are independent from upscaling of method (and location) specific geometric means reported for MADE (Figure 4). The deterministic conductivity conceptualization is suitable for properly modelling the regional groundwater in line with the model aim "Mean Arrival" as specified in section 2.1.

### 3.2.2 Inclusion Structure in Zones (A+B)

Flowmeter logs from MADE show a significant discontinuous heterogeneity in the layering (Figure 2). We represent these structures making use of the binary inclusion structured described in section 2.2. We assume little to no information on horizontal structures and connectivity to mimic typical field situations – thereby deliberately ignoring the large amount of data at MADE. We make use of solely four flowmeter logs (Figure 2a).

The binary conductivity distribution is constructed for the entire domain comprising both deterministic zones. The upstream zone $Z_1$ consists of a bulk of low conductivity $K_1$ with a percentage $p$ of high conductivity $K_2$ inclusions; the downstream zone $Z_2$ vice versa (Figure 5).

We identify the specific values of $K_1$ and $K_2$ from the statistical relationship for binary structures (Rubin, 1995): $\ln \bar{K}_{Z1} = (1-p) \cdot \ln K_1 + p \cdot \ln K_2$ and $\ln \bar{K}_{Z2} = p \cdot \ln K_1 + (1-p) \cdot \ln K_2$ using the mean conductivities of the zones $\bar{K}_{Z1} = 2e-6$ m/s and $\bar{K}_{Z2} = 2e-4$. $p$ is deduced from the flowmeter profiles (Figure 2a). Being from both zones $Z_1$ and $Z_2$, the profiles differ significantly in their average value. However, all show a tendencies of binary behavior with a significant spread over depth. The data is grouped into high and low values being at least two orders of magnitude apart. Then, $p$ is the fraction of values in the minor group, which is $10-20\%$ for the MADE flowmeter profiles (Figure 2a) leading to $p = 15\%$ as default value.

The inclusions structure in both zones is designed according to the simplified block structure outlined in paragraph 2.2. The domain is divided into horizontal blocks of length $I_h$. Each block contains randomly located inclusions of thickness $I_v$. The flowmeter logs at MADE indicate a change in vertical layering every $0.25 - 1$ m (Figure 2a). Thus, we chose $I_v = 0.5$ m. In combination with a inclusion percentage of $p = 15\%$ and an aquifer thickness of $10$ m this gives three inclusions per block.

The parameter $I_h$ is the most difficult to extract from data, due to the limited amount of information on horizontal structures and connectivity. We specify $I_h$ through a pragmatic, but stochastic meaningful approach by combining estimates with parametric uncertainty to rely on as little data as possible: A first guess results from auxiliary data analysis: An anisotropy ratio of $e = 0.1 - 0.025$ is given from the large scale pumping tests (Boggs et al., 1990)). Combining it with the inclusion thickness of $I_v = 0.5$ m gives a range of $I_h \in [5\,m, 20\,m]$. To cover parametric uncertainty we use three different values of $I_h$, namely $5$ m, $10$ m and $20$ m instead of only one. The different inclusion lengths produce distinct effects on connected pathways and thus on the mass distribution. A combined ensemble integrates the character of each inclusion lengths. Figure 5b shows an example structure for $I_h = 10$ m. Note that inclusion can touch, so some inclusions are thicker (e.g. $2I_v = 1$ m) and longer (e.g. $2I_h = 20$ m).

For the Monte Carlo Approach, we create ensembles of 600 individual random realizations, with 200 realizations of each inclusion length $I_h$, while all other parameters are fixed. Preliminary investigations showed that 200 realizations are sufficient to ensure ensembles convergence. Reported flow and transport results for the inclusion structure in zones (A+B) are ensemble means.

### 3.2.3 Sub-scale Heterogeneity in Zones (A+B+C)

We combine modules (A), (B), and (C) to an inclusion structure in deterministic zones with small-scale fluctuations (A+B+C), depicted in Figure 5, bottom. Structural aspects of modules (A) and (B) are the same as described before, including parametric uncertainty for the inclusion length $I_h \in \{5, 10, 20\}$ m. Module C is integrated as log-normal distributed conductivity fluctuations (section 2.2). The characterizing parameters for Module (C) depend on the statistics of the super-ordinate modules (A) and (B).

The log-normal fluctuations $\ln Y(\boldsymbol{x})$ are generated using *gstools* (Müller and Schüler, 2019) with zero mean, since the overall mean conductivity refers to $\bar{K}_{Z1}$ and $\bar{K}_{Z2}$ of the deterministic zones. The log-conductivity variance $\sigma_Y^2$ follows from the "variance gap", as difference between the variance of the inclusion structure and the overall variance. The binary inclusions for the chosen setting have a variance of $\sigma_Z^2 = 5.52$ resulting from $\sigma_Z^2 = p \cdot (1-p) \cdot (\ln K_1 - \ln K_2)^2$ (Rubin, 1995). With an overall variance of $\sigma_F^2 = 5.9$ as indicated by (Bohling et al., 2016) (Figure 4), we arrive at a fluctuation variance of $\sigma_Y^2 \approx 0.5$. We apply an exponential co-variance function with length scale parameters being a fraction of the inclusion length scales: $\lambda_h = 1/4 I_h$ and $\lambda_v = 1/4 I_v$. Testing several ratios, we saw that its impact on transport behavior is negligible. Ensembles consist of 600 realizations.

## 3.3 Numerical Model Settings

Flow and transport are calculated making use of the finite element solver OpenGeoSys (Kolditz et al., 2012) in the ogs5py python framework (Müller et al., 2020). The simulation domain is a 2D cross section within $x \in [-20, 200]$ m and $z \in [52, 62]$ m generously comprising the area of the MADE-1 tracer experiment (Boggs et al., 1992). We applied constant head boundary conditions at the left and right margin with a mean had gradient of $J = 0.003$. Tracer is injected at a well located at $x = 0$ with a central screen of $0.6$ m depth. Injection takes place over a period of $48.5$ h with an injection rate of $Q_{\text{in}} = 1.166e - 5$ m$^3$/s according to the initial conditions reported by Boggs et al. (1992). We use a flux related injection representing natural conditions. For technical details, the reader is referred to the *Supporting Information*.

We checked the impact of dimensionality. A detailed discussion is provided in the *Supporting Information*. We found almost no differences between 2D and 3D simulation setups where the binary structure (Module B) dominates. Extending the binary structure in the horizontal direction perpendicular to main flow does not provide additional degrees of freedom for the flow. Thus, extending the model hardly impacts the flow and thus transport pattern, while significantly increasing computational effort. However, dimensionality effects hold for conductivity conceptualization with prevailing log-normal distribution, i.e. dominated by Module C. The option of complexity reduction by using 2D instead of 3D models is warranted for this application by the fact that conductivity conceptualizations is dominated by the binary structure (module B).

Simulation results are processed like the MADE-1 experimental data. Longitudinal mass distributions are vertical averages and accumulated horizontally over $10$ m slices. Note that the simulated distributions show a full mass recovery. Besides spatial mass distributions for the six times where experimental data is available, we present the break through curves (BTCs) as temporal mass evolution at critical distances, although no BTCs data is reported for the MADE-1 experiment.

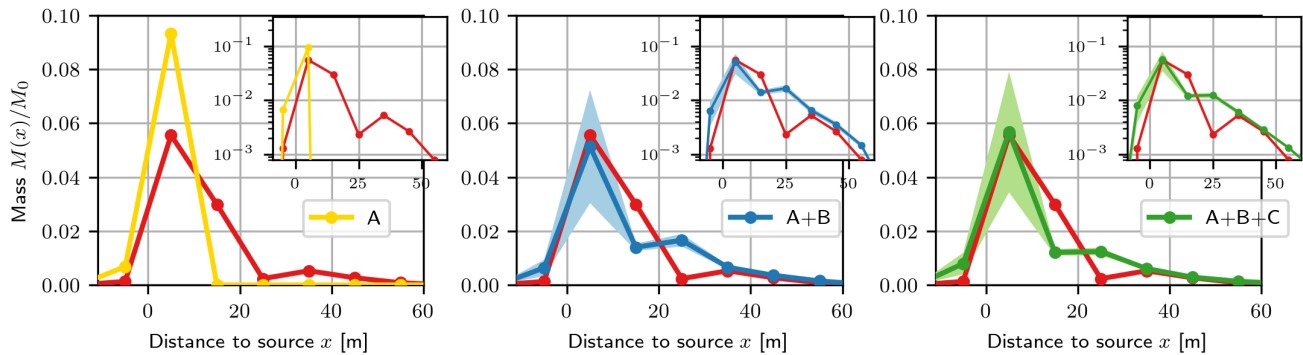

**Figure 6.** Longitudinal mass distribution at $T = 126$ days for conductivity concepts: (A) deterministic zones, (A+B) inclusions in zones, (A+B+C) inclusion in zones with sub-scale heterogeneity (Figure 5). Shaded areas (light blue and green) indicate parametric uncertainty bands. Mass distribution observed at MADE experiment in red. Linear scale and log-scale in subplot.

## 3.4 Simulation Results

Figure 6 shows the simulated longitudinal mass distributions $M(x)/M_0$ of the specified conductivity conceptualizations (section 3.2) at $T = 126$ days after injection. They are compared to the MADE-1 experiment data, which had a mass recovery of $99\%$ at that time.

The mass distribution for the deterministic structure (concept A, yellow) shows a sharp peak close to the injection location and no mass downstream. The conductivity structures with inclusions in deterministic zones (A+B, blue) and with sub-scale heterogeneity (A+B+C, green) result in skewed mass distributions with a peak close to the injection area and a small amount of mass ahead of the bulk. Shaded areas indicate parametric uncertainty due to the variable inclusion length $I_h$. The shade area margins refer to $\pm 3$ ensemble standard deviations, which is similar to the $99\%$ confidence intervals, considering a Gaussian

distribution of variations.

A direct comparison of the mass distributions $M(x)/M_0$ for the structures are depicted in Figure 7 for six temporal snapshots, including $T = 1000$d, where no experimental data is available. The general form of the mass distributions is persistent in time for all conductivity structures.

Figure 8 shows simulated breakthrough curves (BTCs) for the deterministic block and inclusion conductivity structure at

three distances to the injection location. The results for concept (A+B+C) are very close to those of concept (A+B), thus not displayed. Apparent differences to the longitudinal mass distributions as in Figure 7 are due to the spatial data aggregation. The BTC for Module A has the expected Gaussian shape with a late breakthrough at $x = 5$ m given the very low conductivity in the injection area. The stochastic models have an earlier breakthrough and strong tailing at all distances.

BTCs are not available for the MADE-1 transport experiment. However, we added the aggregated mass values at the three

locations for the six reported times in a subplot to indicate a trend of temporal mass development. Note that mass values of the btcs and those at MADE are at different scales due to data aggregation and mass recovery.

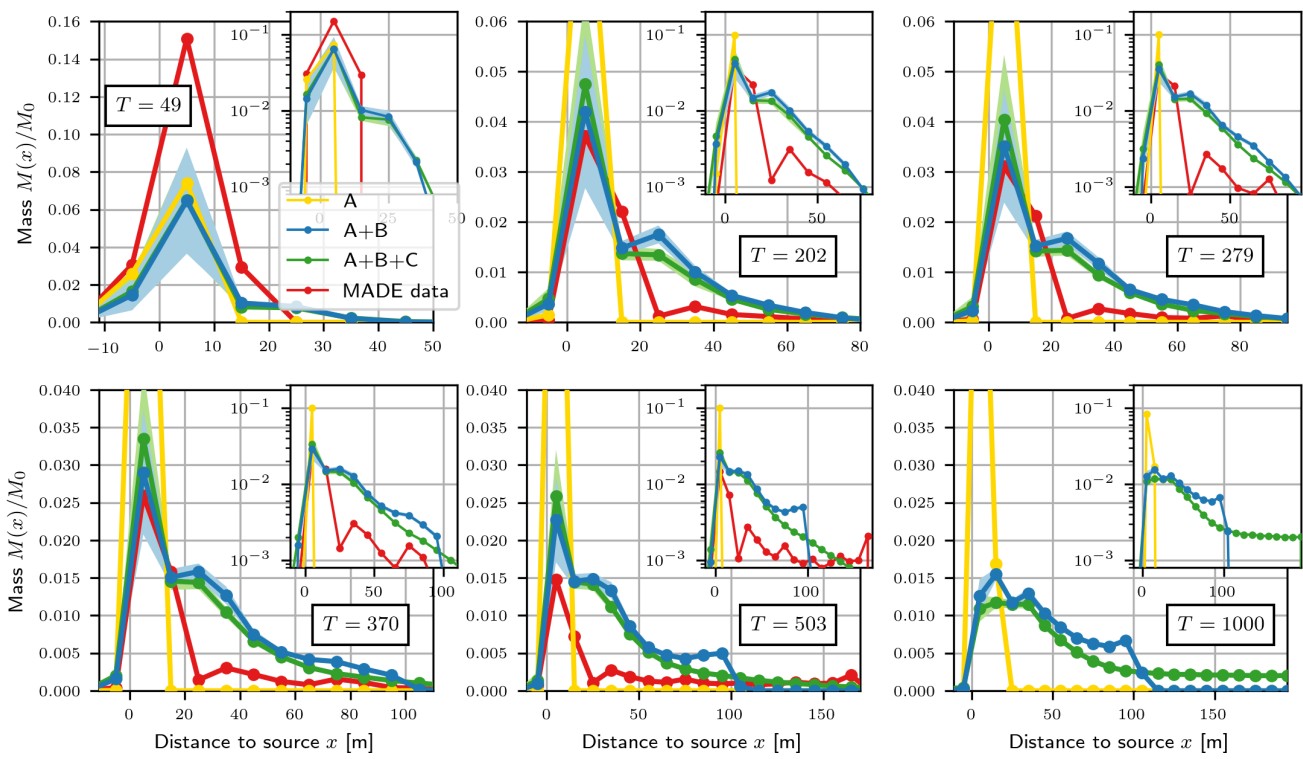

**Figure 7.** Mass distributions at times $T = 49, 202, 279, 370, 503,$ and $1000$ days (panels): red = MADE-1 experiment; yellow = concept (A); blue = concept (A+B); green = concept (A+B+C). Shaded areas (light blue and green) indicate parametric uncertainty bands; semi-log scale in subplot.

## 3.5 Discussion

All conductivity structures were able to reproduce the skewed hydraulic head distribution as observed at MADE (Figure 1a). The corresponding mean flow velocity determines the travel time. As a results, all models properly reproduced the spatial position of the mass peak (Figure 6).

The deterministic block structure (A) failed to reproduce the skewed mass distribution observed at MADE. The leading front mass traveling through fast flow channels could not be predicted (Figure 7) solely using average $K$ values in zones. In line with model aim "Mean Arrival" (section 2.1), the simple structure allows to estimate the regional groundwater movement and to predict the location of the bulk mass. However, in case of aiming at "Risk Assessment", the arrival times of mass would be significantly underestimated, as clearly be observable comparing BTCs (Figure 8).

Tracer transport in a binary conductivity structure with inclusions (concept A+B) reproduces the observed mass, both for the peak near the injection site and the leading front. The simulated longitudinal mass distribution shows a second peak downstream

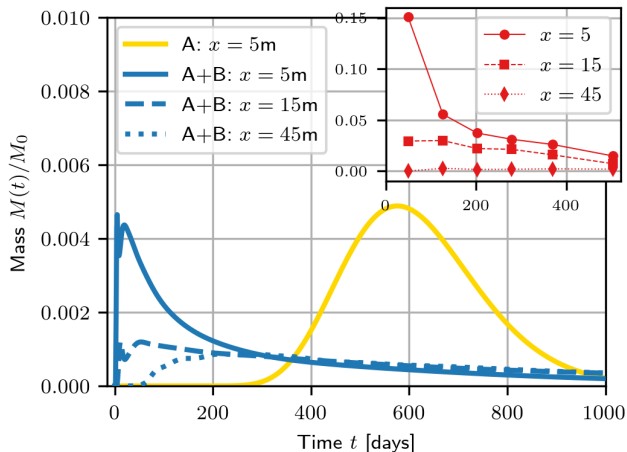

**Figure 8.** Breakthrough curves: Total mass $M(t)/M_0$ versus time at selected control plane locations for inclusion structure (A+B), (blue) at $x = 5\,\mathrm{m}$ (solid), $x = 15\,\mathrm{m}$ (dashed), $x = 45\,\mathrm{m}$ (dotted); and for deterministic structure (A) at $x = 5\,\mathrm{m}$ (solid yellow line). Reported mass values for MADE at the three locations (red markers) given in subplot. Regard the difference in scale due to the spatial averaging of experimental data.

(Figure 7), which increases with time. The position is related to the interface between the low and high conductivity zones at $20\,\mathrm{m}$ distance to the source. Such a second peak is absent in the observed MADE-plume, however it might be associated
with the mass loss for the later times. The skewed mass distribution is related to significantly smaller first arrival times as can be seen for the BTCs in Figure 8 compared to the deterministic structure. The BTCs are clearly non-Gaussian with heavy tailing. It shows the same temporal as the MADE experiment data.

The horizontal inclusion length $I_h$ for structure (A+B) was not fixed, but was varied over the range of $I_h \in \{5, 10, 20\}$ m. The uncertainty bands in Figure 6b indicate that $I_h$ mostly influences the height of the mass peak close to the source. $I_h$
characterized the connectivity of the source area $Z_1$ to the high conductivity zone $Z_2$. Thus, it determines the distance of the bulk mass being trapped in the low conductivity area. The larger $I_h$ the higher is the amount of mass transported downstream. The shape of the leading front is less impacted by $I_h$ giving that its value does not influence the effect of the inclusions as preferential flow per se.

The predicted plume shape for the conductivity structure with inclusions and subscale heterogeneity (A+B+C) is almost
similar to the one without sub-scale heterogneity (A+B). Consequently, the inclusion structure is the one which determines the shape of the distribution, whereas the impact of sub-scale heterogeneity is minor. Given the model aim of plume prediction, the additional effort for determining characterising geostatistical parameters for the sub-scale heterogeneity is not warranted.

The binary conductivity conceptualization (A+B) was derived for MADE with few observations from standard methods, as can be expected to be present at many field sites. The price for the limited amount of data is parametric uncertainty. A

sensitivity study revealed that the mass distribution resulting from the binary conductivity structure is very robust against the choice of parameters. The inclusion length $I_h$ and the choice of the $K$ contrast between the zones show the highest impact. The latter was expected as the mean conductivity determines the average flow velocity and by that the peak location and the general distribution shape. The impact of $I_h$ is represented in the uncertainty bands (Figures 6b, 7). Other parameters as amount of inclusion $p$ and sub-scale heterogeneity parameters as the variance have minor effects. For details, the reader is referred to the

*Supporting Information*. In this regard, the binary structure is very stable towards parametric uncertainty.

## 4   Summary and Conclusions

We introduce a modular concept of heterogeneous hydraulic conductivity for predictive modeling of field scale subsurface flow and transport. Central idea is to combine deterministic structures with simple stochastic approaches to rely on few measurements and to forgo calibration. The scale hierarchy of hydraulic conductivity induces three structure modules which represent:

(A) deterministic large scale features like facies; (B) intermediate scale heterogeneity like preferential pathways or low conductivity inclusions; (C) small-scale random fluctuations. Field evidence of heterogeneity features and module's input parameters are provided by observation methods with the appropriate detection scale. The specific form of the scale-dependent features depends on the site characteristics and field data. We propose a deterministic model for large-scale features, a simple binary statistical model for intermediate and a log-normal random model for small-scale features. However, the integration of alterna-

tive conductivity structures is possible. Thereby, the concept is easily adaptable to any field site making aquifer heterogeneity accessible for practical applications.

An illustrative example is given for the heterogeneous MADE site. Three modular conductivity structures are constructed, based on two observations: (i) the existence of distinct zones of mean flow velocity, and (ii) high conductivity contrasts in depth profiles suggesting local inclusions acting as fast flow channels. The structures are used in a predictive flow and transport

model which is free of calibration. The comparison of results to the MADE-1 field tracer experiment showed that all conceptualizations can be of value depending on the modelling aim. However, predicting the mass plume behaviour required to take heterogeneity into account.

The combination of deterministic and simple binary stochastic showed the best results given the trade-off between transport prediction and need for measurements. Realizations of hydraulic conductivity composed of binary inclusions in two blocks

with different average conductivity. Details on the topology are thereby secondary, since binary structures show robustness towards the choice of specific parameters.

The simple binary structure was able to capture the overall characteristics of the MADE tracer plume with reasonable accuracy requiring only a small amount of observations. Among the few predictive transport models for the MADE site, the presented approach shows a higher level of simulation effort due to the Monte Carlo simulations. However, the lower

level of data requirements makes it attractive for application at less investigated sites. Note that when applying the proposed heterogeneity conceptualization in other modelling application, a 3D model setup should be considered first, particular when heterogeneity is conceptualized by a log-normal distribution (modules C). A complexity reduction to 2D models is warranted

when the heterogeneous conductivity conceptualizations does not impact the flow pattern in transverse horizontal direction, such as the binary structure. The generality of the binary concept makes it easily transferable to other sites; particularly when focusing on a few, but scale-related measurements.

A hierarchical conductivity structure allows to balance between complexity and available data. Large scale structures determine the mean flow behavior, which is most critical for flow predictions. They can be integrated to a model with reasonable low effort. Structural complexity increases with decreasing heterogeneity scale where small-scale features have the highest demand on observation data. However, even with limited information on the conductivity structure, simple stochastic modules can be used to incorporate the effect of heterogeneity. Considering small scale feature, the conductivity structure can be extended by including modules when additional measurements are available.

Distinguishing the effects of the scale-specific features on flow and transport also allows to identify the need for further field investigations and potential strategies. The adaptive construction based on scale-specific modules allows to create a conductivity structure model as complex as necessary but as simple as possible.

The use of simple binary models is very powerful when dealing with strongly heterogeneous aquifers. They require less observation data compared to uni-modal heterogeneity models, as log-normal conductivity with high variances. Binary models also allow to incorporate effects of dual-domain transport models without the drawback of having non-measurable input parameters which require model calibration. Our work shows that highly skewed solute plumes can be reproduced with classical ADE models by incorporating deterministic contrasts and effects of connectivity statistically. In summary, we conclude:

– Modular concepts of conductivity structure allow to separate the multiple scales of heterogeneity. Scale related investigation methods provide field evidence and conductivity model parameters. A hierarchical approach for conductivity can thus reduce observation effort by focusing on the model aim.

– Site specific heterogeneous hydraulic conductivity can be easily constructed with simple methods taking the (limited) amount of data into account. For aquifers with high conductivity contrast, we recommend combining large-scale deterministic structures and simple binary stochastic models.

– The application example at MADE showed that complex field structures can be represented appropriately for transport predictions with an economic use of investigation data.

This work aims to contribute to bridging the gap between the advanced research in stochastic hydrogeology and its limited use by practitioners, being a subject of recent debate (e.g. Rajaram (2016)). We advocate the use of heterogeneity in transport models for successfully predicting solute behavior, particularly in complex aquifers. This can be done with few data and simple tools: adaptive structures allowing to combine deterministic, random binary and geostatistical models depending on the available data and the site-specific modelling aim.

*Code and data availability.* Study related python scripts are public available at *https://github.com/GeoStat-Examples/Binary_Inclusions* (Zech and Müller, 2020), including scrips for (i) generating and (ii) visualizing binary inclusion structures as well as (iii) scripts for transport

simulations in the random inclusion structure adapted to the MADE-1 site settings. The python API *ogs5py* Müller (2019) and the geostatistics packages *gstools* Müller and Schüler (2019) used in this study are both available on *https://github.com/GeoStat-Framework*. Data on the MADE aquifer can be accessed via the stated literature sources. Data generated for this study is available upon request to the corresponding author.

*Author contributions.* All authors contributed to developing the approach and writing the paper. Simulations and figure preparation was
performed by AZ.

*Competing interests.* No competing interests.

*Acknowledgements.* The authors like to thank Jeff Bohling for background information on Flowmeter and DPIL data. We thank the editor and reviewers for their helpful comments.

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
