# Peer review of "A Field Evidence Model: How to Predict Transport in Heterogeneous Aquifers at Low Investigation Level?"

_Hydrology and Earth System Sciences, 2020_

## Short Comment (SC1) · 27 Feb 2020

This article presents an approach re 'A hierarchical aquifer model which combines large-scale deterministic structures and simple stochastic approaches' in order to 'Predict Transport in a Heterogeneous Aquifers'

As it is a research paper we may expect this to be a novel approach. If the paper is extending similar earlier work then references should be made. If it is routine application of existing methods the paper should be called a case history.

The paper presented here shows neither a novel nor an original approach. The ap-

proach presented was also earlier applied to the same site where the MADE project was conducted (Columbus Air Force Base, MS, USA)

The type of hierarchical deterministic/stochastic modelling of geological features and permeability distribution discussed in the paper, has been extensively used in the oil and gas industry since the mid 1980s. There is a vast body of literature on the methodology and applications. All this is completely ignored, ie. not referenced, in this paper. Plenty basic references (up to 1996) can be found in chapter 2 of ref 1 below.

This type of model is also not new for the Columbus Air Force Base area where the MADE experiment was conducted. I have personally published a PhD thesis and an article on a hierarchical deterministic/stochastic approach applied to tracer tests at Columbus Air Force Base (the site where the MADE experiment was conducted). The 4th listed author is well aware of all this, as he personally communicated with me, was reviewer of my PhD thesis (Ref 1 below), and attended conferences where papers were presented (eg. ref 2)

Given this, the authors should thoroughly re-study existing literature and reference some key papers out of the oil and gas industry. They also should make very clear that this is not a novel/original approach but simply a standard application of what has done before and is routine in oil and gas reservoir modelling. The authors should also make clear reference to similar work already conducted $\sim$ 25 years ago at the same site (Columbus Air Force Base test site where the MADE experiment was conducted), eg. ref 2.

The only reason why the material could be published, is that it finally may point out the scientific confusion and structural research mis-management around the MADE experiment and stochastic hydrology. The MADE experiment has led to numerous publications in journals, which all ignored to account for geological heterogeneity in an appropriate manner and ignored other work that would not fit the premises of stochastic hydrology (macro dispersion theory).

Ref 1 - Herweijer, J.C., 1997. Sedimentary heterogeneity and flow towards a well. Ph.D. dissertation, Free University, Amsterdam (https://www.hydrology.nl/images/docs/dutch/1997.01.07_Herweijer.pdf)

Ref 2 - Herweijer, J.C,1996. Use of sedimentology andgeostatistical modeling to estimate uncertainty of groundwater models. Proc.International Conference on Calibration and Reliability in Groundwater modeling(ModelCARE 96), Golden (CO, USA), September, 1996 (https://pdfs.semanticscholar.org/a5a5/25d8da8091bb59a59795262d932f0b4a6333.pdf)

Please also note the supplement to this comment:
https://www.hydrol-earth-syst-sci-discuss.net/hess-2020-30/hess-2020-30-SC1-supplement.pdf

**Supplement:**

[supplement omitted: unrelated document]

---

## Referee Comment (RC1) · Anonymous Referee #1 · 9 Mar 2020

The manuscript presents a hierarchical approach for modeling flow and transport in heterogeneous aquifer. The approach is applied to the now classic MADE macrodispersion experiment, and it is focused on the modeling of longitudinal mass distribution, as observed during the course of the experiment. The paper is very well written and the method is clearly illustrated. The topic is relevant, and I do believe that approaches like the one envisioned here are very important to reduce the complexity of natural groundwater systems. I think that the work deserves publication. A few minor comments follow.

- Abstract: I find unusual to start new paragraphs within an abstract.

[Figure]

- Eq.1: why the ADE is presented in one spatial dimension? This may be misleading, also considering that hydraulic conductivity K(x) is variable in x only under such conditions (line 32)

- Figure 1. The position of the boundary between orange and green is not clear: where does it derive?

- Line 155: is there any evidence of such large differences in hydraulic conductivity among zones (several orders of magnitude, line 157)?

- Line 164: "reproduce" instead of "reprocude"

- Line 175: why the choice of Ih and Iv?

- Line 181: please elaborate more on the "expert knowledge" for assessing the range of reasonable Ih values estimated

- Line 189: please define ergodicity, and briefly explain (possibly with references) why it is assumed when the plume has travelled 10-100 characteristics lengths.

- Line 248: I don't see a clear transition at x=20 from Fig.1

- Line 250: I don't recall the contrast described here in Boggs et al (1992), please elaborate more

- Line 257: "designed" instead of "design"

- Line 272: please explain the heuristic approach with some more detail.

- Line 275: why 600 realizations?

- Line 294: so the model is 2d? why not working with the more realistic 3d setup? Do you expect differences in the results? I guess that the additional degree of freedom brought by 3d could make a difference.

- Line 296: how is the solute injected? Does the local injection rate depend on local hydraulic conductivity?

- Figure 6. Please introduce a legend.

- Conclusions: the first item of the list (line 396) is a rather well known and general statement, I would not add it as one of the conclusive statements of the work.

---

## Short Comment (SC2) · 9 Mar 2020

2D vs 3D has been looked extensively, see e.g.

Static characterizations of reservoirs: refining the concepts of connectivity and continuity Joseph M. Hovadik and David K. Larue Petroleum Geoscience, Vol. 13 2007, pp. 195–211

http://citeseerx.ist.psu.edu/viewdoc/download?doi=10.1.1.818.7201&rep=rep1&type=pdf

and

King, P. R., 1990, The connectivity and conductivity of overlapping sand bodies. In,

[Figure]

Buller Anthony T. et al.,eds, North Sea oil and gas reservoirs; II, Proceedings of the North Sea oil and gas reservoirs conference. [Book, Conference Document] Pages 353-362.

https://link.springer.com/chapter/10.1007/978-94-009-0791-1_30

Models in study @ MADE site mentioned in my previous comments were all run in 3D, as is was concluded that 2D models tend to 'suppress' connectivity.

---

## Referee Comment (RC2) · Anonymous Referee #2 · 23 Mar 2020

The manuscript presents a hierarchical approach for modeling flow and transport in heterogeneous aquifer. The key idea is combining large-scale deterministic structures and simple stochastic approaches. While the inclusion of a hierarchical structure to deal with heterogeneous structure is not new (some modelers have used similar ideas, yet not as structured as in this case), the authors introduce a formalism to make it understandable and efficient, I think is the main value of the manuscript. A significant point in the manuscript is the n-th try to model the data from the MADE side (here n is a very, very large number).

So, maybe the main comment I have is the issue of dimensionality. First, the very

simple thing is that eq. (1) should be 3D, as this is a general idea, and no need to simplify the problem at this point (you can do that later). But, most importantly, your application is 1D. I have seen many models trying to fit the 1D data of MADE; but after all these years, I have not yet seen the spatial distribution of values. Everybody reports the correspondence with transects (your figures 6 and 7). Transects are ok, but do not reflect the real picture at all. From l.235, "Concentrations were observed within a spatially dense monitoring network at several times after injection". Is this data available? Why nobody uses it in their models? You start with Figure 1. Why such a simple concept, if we know that it is slightly more complicated.

But, in general I like the work, and I feel it is very well written. I loved in particular the section "Exemplary Model Aims". This is written in a very didactic way.

This is a tough one and I do not expect an answer. The model developed in Section 3.2 involves quite a number of decisions and parameters. Then you get a reasonable fit. Now, can you really calibrate this model with so many parameters at very different scales (variances, integral distances, p values, anisotropy ratios, directions of anisotropy,...)? I can see that being done manually for one-two parameters (e.g., your line 276), but more? You would need a supercomputer and plenty of staff or students working on it, but this would be a waste. So, is there any automatic calibration approach that you envision in the future?

Minor issues:

The problem inherent to hierarchy of scales is how do you assign variability to one scale or the other one. I mean, you can always claim that some general trends are nothing but randomness if we look at a larger scale. Some discussion about how to distinguish Modules (A), (B) and (C) in a general case could benefit the paper. I mean, should (B) always related to the transport features as suggested in l.166?

You could comment a bit on recharge, because probably recharge and transmissivity (in a 2D scenario) might be correlated. How does your hierarchy approach deal with

this parameter? Similarly, you could also comment a bit on the impact of porosity, if you think it is relevant (maybe it is not); it appears in the transport equation.

L79. In my opinion the models of Fiori (2013, 2017) are completely non-predictive (actually, they are based on wrong assumptions, as you show in your paper); outperforming those methods should not even be cited.

L90. Again, the use of word "macro-dispersion"; maybe you refer to "enhanced dispersion". The concept of "macro" refers to a specific quantity (since the original derivation of Gelhar and Axness, to all those by Dagan and so) that are never, ever, attained in real field conditions.

L116. Is this reference really needed here? I mean, the relevance of pumping tests comes from the 1930's if not earlier. And we teach them in class. . .

L 254. "Arrival" is misspelled

L 312. This is equivalent only if a gaussian distribution of concentrations is invoked. You could add this warning.

---

## Author Comment (AC1) · 7 Apr 2020

We want to thank Joost Herweijer for his interest in our work. The points he raised will be addressed critically during the revision process of the manuscript in addition to the comments of the reviewers. We acknowledge the initiation of the scientific discussion on the subject matter of integrating aquifer heterogeneity to hydrogeological transport models as this was one of the goals of the work. As stated in the manuscripst's last paragraph, we aim to contribute to bridging the gap between the advanced research in stochastic hydrogeology and its limited use by practitioners. In this line, we agree that hierarchical deterministic/stochastic modeling permeability is used in the oil and gas

industry, but hardly found it's way into applied hydrogeology.

We acknowledge his effort in providing us publications on MADE which have not been available to us or of which we were not aware, respectively. Missing literature references will be integrated to the paper. As part of the updating process, we will discuss similarities and differences between the mentioned work of J. Herweijer on the MADE site (referenced PhD thesis) and ours. In the same line, we also discovered that other references are missing in the current version, as e.g. the inspiring work of Bianchi & Zheng, 2016 on the MADE-2 experiment.

---

## Author Comment (AC2) · 18 May 2020

First, we want to thank the referee for his positive evaluation of our work. We appreciate the time and effort he put into reviewing the manuscript. The paper will benefit from revising it according to the comments.

This response will address all general questions raised. Consequential text modifications in the manuscript will be outlined in the next step of the reviewing process (not intended in this step) along with correction of typos etc. Referee's points referring to the same topic, are bundled giving a deviation of the original order at times.

- Eq. 1 will be adapted to 3D.

- The boundary and contrast between the areas of distinct hydraulic properties: (raised in relation to Figure 1, Ls 155, 248, and 250):

  – The position of the boundary between blocks: Studying Figure 1 (left) shows that 20m down stream of the source (black dot) the head pattern changes abruptly. Along the orange line in the left figure there are 4 head isolines whereas along the green line of 40m length only one. This is a strong indication for a change of mean hydraulic conductivity. Thus, we chose this position as location for the interface of distinct material blocks. Figure 1 (right) indicates the vertical cross section where the choice of the coordinate system is along the one outline in the left figure. Part of this explanation will be integrated to the manuscript to clarify the choice and figures.

  – evidence of these large differences, e.g. by two large scale pumping tests as discussed in e.g. Boggs et al., 1992. We will add the reference at that location and elaborate.

- (L. 174ff) Inclusion topology: We will revise the paragraphs on the inclusion topology according to the points raised by the reviewer, explaining the choice of $I_h$ and $I_v$ and elaborate on what we mean with "expert knowledge".

- (L 189) We can extend the text to give a definition of ergodicity: Intuitively speaking, the ergodic hypothesis for a system implies that all states of the ensemble are available in each realization [Dagan, 1989]. A figurative description in the context of transport is, that the plume sampled sufficient heterogeneity over its travel distance to be representative for the average behavior of the heterogeneous material structure. The value of 10-100 characteristics lengths follows from stochastic arguments of the sample size [Dagan, 1988,1989].

- (L. 272) Inclusion's structure and choice of horizontal inclusion length $I_h$:

- The parameter $I_h$ is the most difficult to extract from data, generally due to the very limited amount of information on horizontal structures and connectivity. Thus, a pragmatic, but also stochastic meaningful approach is necessary. We decided to combine estimates from the data (the range of $I_h \in [5m, 20m]$ deduced from vertical inclusion length and the anisotropy rate), with the approach of parametric uncertainty: instead of using only one value out of the range, we allow for 3 different: 5m, 10m and 20m. The different inclusion length produce distinct effects on connected pathways and thus on the mass distribution. In the combined ensemble the character of each inclusion length is thus integrated.
- The ensemble thus consists of 3* 200 realization of each inclusion length.
- The formulation heuristic approach might be misleading. We will consider reformulating it. We will also expand the paragraph providing the additional information outlined above.

- (L 275): We used 600 realizations to assure that the number is sufficiently large to ensure ensemble convergence. As stated in the manuscript, we found in preliminary convergence tests, that 200 realizations are sufficient to reproduce ensembles averages. Given the combination of different inclusion length (previous comment), we combined 3*200 realizations for the general ensemble representing model structure A+B.

- (L. 294): Dimensionality of the model: The model is indeed 2D not 3D.

  - In preparation of the model, we also performed 3D simulations (for a reduced set of realization due to computational effort) and found almost no differences to 2D results. This can be explained by the conceptualization of the heterogeneous binary structure. So addressing the question "You expect differences in the results?" - Almost non; nor in the flow pattern and hence in the mass distribution pattern. Thus, we decided to stick with 2D.

– Extending the binary structure in the y-direction will be like combining many copies of the 2D cross section perpendicular to flow. This will cause no real change in the flow pattern. The inclusion length in y-direction is the same as in x-direction (as long as there is no indication of horizontal anisotropy). Thus, the binary structure does not change over several meters in y-direction. Inclusions extend along and perpendicular to the flow direction, giving the flow no reason to deviate from the main flow path. In this sense, the flow pattern is hardly impacted by the additional degree of freedom and the mean flow velocity is almost identical in 2D and 3D.

– We are aware that this is in contrast to log-normal random fields, where flow in uniform fields shows higher effective K values in 3D than 2D. The difference is: in log-normal fields, K-values change gradually in all directions. Thus, adding a 3rd dimension perpendicular to the main flow direction allows to circumvent areas of low conductivity and thus increases the effective mean flow velocity. In the binary material, there are no gradual changes. A layer of low conductivity in horizontal direction extends in both, x- and y-direction, being an obstacle for the flow and not allowing for "flowing around" in y-direction.

– The very light differences between 2D and 3D we observed, we refer to a slight increase of mean flow velocity due to a higher connectivity of inclusion in 3D. However, this is only relevant over a large domain and does hardly impact the local flow pattern in the area where transport takes place.

– A 3D model would be a more realistic setup, but in this particular application it does not change the results due to the conceptualization of the binary model. The additional degree of freedom does not impact the flow pattern. Thus, a 3D model increases effort but brings no benefits. In contrary, setting up Monte Carlo simulation with a 3D model would keep practitioners from adapting our approach for other field situation. However, we agree that for other conceptualization of heterogeneous hydraulic conductivity a 3D model

    setup is preferable.

- (L. 296): Solute injection follows the experimental description in Boggs et al., 1992. It is a flux related injection being the realistic representation of natural conditions. Thus the local distribution of tracer depends o the local heterogeneity.

- (Figure 6): We will add a legend to Figure 6.

- (L. 396) We will adapt the conclusions accordingly.

---

## Author Comment (AC3) · 18 May 2020

We'd like to thank the referee for taking his time to review our manuscript and appreciate his positive evaluation and constructive criticism. We are going to revise the paper accordingly. The manuscript will improve based on the referee's comments. In this response, we plan to address the questions raised. Consequential text modifications in the manuscript will be outlined in the next step of the reviewing process (not intended in this review step) along with correction of typos etc:

- The the question of dimensionality (of the data):

[Figure]

- – Eq. 1 will be adapted to 3D.

- – First we want to specify that the conceptual and numerical models are 2D. We postprocess the calculated mass distributions to allow a comparison with the 1D reference data of the MADE 1 experiment: As outline in the manuscript, averaged over the directions perpendicular to the flow and aggregated over intervals of 10m.

- – Now, the critical point raised by the reviewer: "After all these years, I have not yet seen the spatial distribution of values." - We neither. Unfortunately, no other mass data than the 1D transects is available to us. In correspondence with many colleagues, we figured that the raw data is not public. We regret this situation, but are not in the position to change it. So, I can just agree to the referee.

- – Addressing the point: "You start with Figure 1. Why such a simple concept, if we know that it is slightly more complicated." - That is actually one of the paper's targets: make use of the "simple concepts" and usually available data as piezometric surface maps to construct a reasonable heterogeneous hydraulic conductivity structure. In the application of the concept to MADE we wanted to show that by integrating and combining "basic" data, it is possible to reproduce apparently complex mass distribution patterns at least at the level of spatially integrated longitudinal mass distributions.

- • The point of parameters to specify for the model:

  - – In section 3.2, we tried to outline how to derive the required parameters from hydraulic observations. We aim to emphasize that the model is set up as a predictive model. There is no calibration involved. This is also the target for application: practitioners should be able to setup a transport model without relying on calibration. Th point of a predictive and calibration-free model will be stronger emphasized in the revised manuscript.

- – The choice of several values of inclusion length (l. 276) is not a calibration but an integration of parametric uncertainty. We did not calibrate the model to one of the values, but included random realization with all of these values to the ensemble.

- – We agree, that the general model contains many parameters at very different scales (variances, integral distances, p values, anisotropy ratios, directions of anisotropy,. . .). However, as outline later, they do not necessarily all be part of a conceptual model setup at a site. The model should be adapted to the available data and local conditions. As can be seen in the application to MADE: Module C (fine scale heterogeneity) does hardly impact the overall mass distribution. Thus, without this module, the number of required parameters is much smaller. However, there might be other application cases, where there is no indication of large contrasts warranting for Module A, thus a conceptual model only consisting of Module C might be apt for some sites, again resulting in a manageable amount of required parameters.

- hierarchy of scales: We fully agree with the referee that categorizing spatial variability observed at a specific site to scales is subject of discussion and uncertainty. We will integrate some discussion on how to distinguish Modules (A), (B) and (C) in a general case into section 2.2.

- We can add a notice on recharge and porosity. However, hydraulic conductivity will be the dominating parameter given the scale at which it varies (order of magnitude) compared to recharge and porosity (in a range of factor 2). Particularly here, where we advocate the use of heterogeneity aquifer structuring due to high conductivity contrasts. In this line, notice the uncertainty associated with hydraulic conductivity observations (differences in mean behavior between methods ranges up to 2 order of magnitude, see Figure 4). Thus, the range of uncertainty induced by hydraulic conductivity will outperform the impact of fluctuations in recharge and porosity.

- (L 79) We cannot agree to the referee's opinion on the work of Fiori et al. (2013, 2017). Being previous modeling work on MADE, we prefer to keep the reference.

- (L 90) In the context here, we agree that macrodispersion is not the proper choice of words. We are going to modify it to "enhanced dispersion". Generally, we use the word "macrodispersion" in the sense as defined by Gelhar and Axness [1983] and the work of Dagan [1986] and following.

- (L 116) We can remove the reference.

- (L 312) We will add the warning.

To all points raised and discussed above, we will integrate comments and text modifications in the manuscript to clarify. Typos are not listed above, but will of course be corrected.

---

## Author Response (AR1)

**Rebuttal**

Dear editor, referees and discussant,

thank you for your constructive comments and interest in our work. We revised the manuscript accordingly. In the following, we provide point-by-point replies (in blue) to the comments (in italic). While line numbers in the comments refer to the previous manuscript version, mentioned lines in the response relate to the revised manuscript. Attached to the response is a marked-up manuscript version with tracked changes.

With kind regards,

Alraune Zech
on behalf of the author-team.

**Editor comment:**

*I would suggest the authors should try to improve a bit the overall scientific significance of their study by highlighting some novel aspects and advances of the present study with respect to the existing literature on the subject.*

We followed the advice of the editor along the comments of the discussant and the referees. We updated the abstract as well as passages of the introduction (l 64 ff) and conclusion section (422ff) accordingly.

**Response to Referee #1:** ()

*The manuscript presents a hierarchical approach for modeling flow and transport in heterogeneous aquifer. The approach is applied to the now classic MADE macrodispersion experiment, and it is focused on the modeling of longitudinal mass distribution, as observed during the course of the experiment. The paper is very well written and the method is clearly illustrated. The topic is relevant, and I do believe that approaches like the one envisioned here are very important to reduce the complexity of natural ground-water systems. I think that the work deserves publication. A few minor comments follow.*

We want to thank the referee for his positive evaluation of our work. We appreciate the referee's time and effort he put into reviewing our manuscript. The paper will benefit from revising it according to his comments. Along the lines of the author comments published before, we here address all raised in combination with text modification in the manuscript.

- *Abstract: I find unusual to start new paragraphs within an abstract.*
  We decided for a structuring of the abstracts into paragraphs to improve readability.

- *Eq.1: why the ADE is presented in one spatial dimension? This may be misleading, also considering that hydraulic conductivity K(x) is variable in x only under such conditions (line 32)*
  Eq. 1 and related quantities are adapted to 3D.

- *Figure 1. The position of the boundary between orange and green is not clear: where does it derive?*
  The position of the boundary between blocks: Studying Figure 1 (left) shows that 20m down stream of the source (black dot) the head pattern changes abruptly. Along the orange line in the left figure there are 4 head isolines whereas along the green line of 40m length only one. This is a strong indication for a change of mean hydraulic conductivity. Thus, we chose this position as location for the interface of distinct material blocks. Figure 1 (right) indicates the vertical cross section where the choice of the coordinate system is along the one outline in the left figure.

  We added a comment in the caption of Figure 1. We expanded the explanation of Figure 1 in the text (section 2.1.1, L.130)

- *Line 155: is there any evidence of such large differences in hydraulic conductivity among zones (several orders of magnitude, line 157)?*
  This section focus on the general setup of Module A. It is designed to represent deterministic areas of high conductivity contrast in the conceptual model for sites with field evidence. We added an explanatory sentence to the paragraph (L. 163).
  For the MADE site, there is evidence of such large differences in hydraulic conductivity (Figure 1 and the explanation on that). Thus, the use of Module A in a conductivity conceptual setup is warranted.

- *Line 164: "reproduce" instead of "reprocude".* Corrected.

- *Line 175: why the choice of Ih and Iv?*
  We corrected the misleading formulation (L. 182ff): Figure 2 provides a visual example. The choice of $I_h$ and $I_v$ is thus arbitrary here. The specific parameters are transferred to the figure caption.

- *Line 181: please elaborate more on the "expert knowledge" for assessing the range of reasonable $I_h$ values estimated.*
  We revised the paragraphs on the inclusion topology elaborating on how to determine reasonable estimates for horizontal inclusion length scales in the context of expert knowledge (L.184ff).

- *Line 189: please define ergodicity, and briefly explain (possibly with references) why it is assumed when the plume has travelled 10-100 characteristics lengths.*
  Intuitively speaking, the ergodic hypothesis for a system implies that all states of the ensemble are available in each realization [Dagan, 1989]. A figurative description in the context of transport is, that the plume sampled sufficient heterogeneity over its travel distance to be representative for the average behavior of the heterogeneous material structure. The value of 10-100 characteristics lengths follows from stochastic arguments of the sample size [Dagan, 1988,1989].

  We revised the paragraph adding a definition of ergodicity and we gave references to the relation to travel distance. (L. 198ff)

- *Line 248: I don't see a clear transition at x=20 from Fig.1*
  The point was emphasized in Figure 1 and section 2.1.1 (see also comment above).

- *Line 250: I don't recall the contrast described here in Boggs et al (1992), please elaborate more.*
  A statement on the conductivity contrast is repeatedly mentioned in the Boggs et al, 1992 paper, starting in the abstract: "This asymmetry was produced by accelerating groundwater flow along the plume travel path that, in turn, resulted from an approximate 2-order-of-magnitude increase in the mean hydraulic conductivity between the near-field and far-field regions of the site."
  In the manuscript's paragraph we state all relevant information and refer to the supporting information for further explanations. We do not know what additional information to provide here.

- *Line 257: "designed" instead of "design".* Corrected.

- *Line 272: please explain the heuristic approach with some more detail. Line 275: why 600 realizations?*
  The parameter $I_h$ is the most difficult to extract from data, generally due to the very limited amount of information on horizontal structures and connectivity. Thus, a pragmatic, but also stochastic meaningful approach is necessary. We decided to combine estimates from the data (the range of $I_h \in$ [5m, 20m] deduced from vertical inclusion length and the anisotropy rate), with the approach of parametric uncertainty: instead of using only one value out of the range, we allow for 3 different: 5m, 10m and 20m. The different inclusion length produce distinct effects on connected pathways and thus on the mass distribution. In the combined ensemble the character of each inclusion length is thus integrated. The ensemble thus consists of 3* 200 realization of each inclusion length.
  We used 600 realizations to assure that the number is sufficiently large to ensure ensemble convergence. As stated in the manuscript, we found in preliminary convergence tests, that 200 realizations are sufficient to reproduce ensembles averages. Given the combination of different inclusion length (previous comment), we combined 3*200 realizations for the general ensemble representing model structure A+B.

  We reworked and expanded the paragraphs on the inclusion structure and the number of realizations accordingly. (L. 298ff) In this line, we modified the formulation heuristic approach which is misleading.

- *Line 294: so the model is 2d? why not working with the more realistic 3d setup? Do you expect differences in the results? I guess that the additional degree of freedom brought by 3d could make a difference.*
  Dimensionality of the model: We provided a detailed discussion in the (previous) author comment to the referee. A paragraph on that is added to the manuscript and we provide a detailed discussion along these lines in the Supporting information.

- *Line 296: how is the solute injected? Does the local injection rate depend on local hydraulic conductivity?*
  Solute injection follows the experimental description in Boggs et al., 1992. It is a flux related injection being the realistic representation of natural conditions. Thus the local distribution of tracer depends o the local heterogeneity.

  We added the explanation ("It is a flux related injection being the realistic representation of natural conditions.") to the manuscript.

- *Figure 6. Please introduce a legend.*
  We modified the text (A, A+b, A+B+C) to a legend and introduced a legend in Figure 7 as well.

- *Conclusions: the first item of the list (line 396) is a rather well known and general statement, I would not add it as one of the conclusive statements of the work.*
  We removed the sentence from the conclusive statements.

**Response to Referee #2:**

()

*The manuscript presents a hierarchical approach for modeling flow and transport in heterogeneous aquifer. The key idea is combining large-scale deterministic structures and simple stochastic approaches. While the inclusion of a hierarchical structure to deal with heterogeneous structure is not new (some modelers have used similar ideas, yet not as structured as in this case), the authors introduce a formalism to make it understandable and efficient, I think is the main value of the manuscript. A significant point in the manuscript is the n-th try to model the data from the MADE side (here n isa very, very large number).*

We'd like to thank the referee for taking his time to review our manuscript and appreciate his positive evaluation and constructive criticism. We revised the manuscript accordingly. Here we outline the changes following the discussion in the final author comments.

The aspect of novelty and that this model is yet another try to model the MADE side tracer test was also raised the a discussant and the editor. We addressed these issues by revising the abstract as well as introduction and conclusion section.

*So, maybe the main comment I have is the issue of dimensionality. First, the very simple thing is that eq. (1) should be 3D, as this is a general idea, and no need to simplify the problem at this point (you can do that later).*
Eq. 1 was adapted to 3D.

*But, most importantly, your application is 1D.*
We want to specify that the conceptual and numerical models are 2D. We post-process the calculated mass distributions to allow a comparison with the 1D reference data of the MADE 1 experiment: As outline in the manuscript, averaged over the directions perpendicular to the flow and aggregated over intervals of 10m. (L. 341).

*I have seen many models trying to fit the 1D data of MADE; but after all these years, I have not yet seen the spatial distribution of values.* - We neither.
*Everybody reports the correspondence with transects (your figures 6 and 7). Transects are OK, but do not reflect the real picture at all. From l.235, "Concentrations were observed within a spatially dense monitoring network at several times after injection". Is this data available? Why nobody uses it in their models?*
Unfortunately, no other mass data than the 1D transects is available to us. In correspondence with many colleagues, we figured that the raw data is not public. We regret this situation, but are not in the position to change it. So, I can just agree to the referee.
We added a comment on the data situation to section 3 (L. 245)

*You start with Figure 1. Why such a simple concept, if we know that it is slightly more complicated.*
That is actually one of the paper's targets: make use of the "simple concepts" and data which is often available such as piezometric surface maps to construct a reasonable heterogeneous hydraulic conductivity structure. In the application of the concept to MADE we wanted to show that by integrating and combining "basic" data, it is possible to reproduce apparently complex mass distribution patterns at least at the level of spatially integrated longitudinal mass distributions. We added a comment on that to section 3 (L. 243).

*But, in general I like the work, and I feel it is very well written. I loved in particular the section "Exemplary Model Aims". This is written in a very didactic way.*

*This is a tough one and I do not expect an answer. The model developed in Section 3.2 involves quite a number of decisions and parameters. Then you get a reasonable fit. Now, can you really calibrate this model with so many parameters at very different scales (variances, integral distances, p values, anisotropy ratios, directions of anisotropy,...)? I can see that being done manually for one-two parameters (e.g.,your line 276), but more? You would need a supercomputer and plenty of staff or students working on it, but this would be a waste. So, is there any automatic calibration approach that you envision in the future?*
In section 3.2, we outlined how to derive the required parameters from hydraulic observations. We emphasize that the model is set up as a predictive model. There is no calibration involved.
We saw the need to emphasized this point (predictive and calibration-free model) in the revised manuscript: as e.g. in L. 83, 238, 404.

The choice of several values of inclusion length is not a calibration but an integration of parametric uncertainty. We did not calibrate the model to one of the values, but included random realization with all of these values to the ensemble.
The paragraph on the derivation of the inclusions structure and choice of horizontal inclusion length $I_h$, was reworked and expanded accordingly (L. 302ff).

*Minor issues:*

*The problem inherent to hierarchy of scales is how do you assign variability to one scale or the other one. I mean, you can always claim that some general trends are nothing but randomness if we look at a larger scale. Some discussion about how to distinguish Modules (A), (B) and (C) in a general case could benefit the paper. I mean,should (B) always related to the transport features as suggested in l.166?*
We fully agree with the referee that categorizing spatial variability observed at a specific site to scales is subject of discussion and uncertainty. Addressing the point "I mean, should (B) always related to the transport features as suggested in l.166?" (L 166: heterogeneous features at the same length scale as the plume transport itself") - Not per se. Generally, we relate the Modules to the typical length scale of material feature, also related to specific observation methods. However, Module (B) represents heterogeneity of a few meters length (up to some tenth of meters), which coincides with the typical length scale of a contaminant plumes. In this sense, Module (B) is prone to be representing the relevant heterogeneity.

We added a section (L. 227ff) discussing the hierarchy of scales and how to distinguish modules.

*You could comment a bit on recharge, because probably recharge and transmissivity (in a 2D scenario) might be correlated. How does your hierarchy approach deal with this parameter?*

*Similarly, you could also comment a bit on the impact of porosity, if you think it is relevant (maybe it is not); it appears in the transport equation.*
We add a notice on recharge and porosity in L. 259.
As a site note: In our model, we do not work with transmissivity since there are variations in conductivity and flow velocities in the vertical direction.

*L79. In my opinion the models of Fiori (2013, 2017) are completely non-predictive (actually, they are based on wrong assumptions, as you show in your paper); outperforming those methods should not even be cited.*
We refrain from removing the reference to Fiori et al. (2013, 2017). They offer an alternative model for the MADE site, which we consider as valuable contribution to the scientific discussion.

*L90. Again, the use of word "macro-dispersion"; maybe you refer to "enhanced dispersion". The concept of "macro" refers to a specific quantity (since the original derivation of Gelhar and Axness, to all those by Dagan and so) that are never, ever, attained in real field conditions.*
In the context here, we agree that macrodispersion is not the proper choice of words. We corrected accordingly ( L. 96).

*L116. Is this reference really needed here? I mean, the relevance of pumping testscomes from the 1930's if not earlier. And we teach them in class…-* We removed the reference.

*L 254. "Arrival" is misspelled. -* Corrected.

*L 312. This is equivalent only if a Gaussian distribution of concentrations is invoked. You could add this warning. -* We added the warning (L. 353).

**Response to Discussant:**

*()*

*This article presents an approach re 'A hierarchical aquifer model which combines large-scale deterministic structures and simple stochastic approaches' in order to 'Predict Transport in a Heterogeneous Aquifers'.*
*As it is a research paper we may expect this to be a novel approach. If the paper is extending similar earlier work then references should be made. If it is routine application of existing methods the paper should be called a case history.*
*The paper presented here shows neither a novel nor an original approach. The approach presented was also earlier applied to the same site where the MADE project was conducted (Columbus Air Force Base, MS, USA).*
*The type of hierarchical deterministic/stochastic modelling of geological features and permeability distribution discussed in the paper, has been extensively used in the oil and gas industry since the mid 1980s. There is a vast body of literature on the methodology and applications. All this is completely ignored, ie. not referenced, in this paper. Plenty basic references (up to 1996) can be found in chapter 2 of ref 1 below.*
*This type of model is also not new for the Columbus Air Force Base area where the MADE experiment was conducted. I have personally published a PhD thesis and an article on a hierarchical deterministic/stochastic approach applied to tracer tests at Columbus Air Force Base (the site where the MADE experiment was conducted). The 4th listed author is well aware of all this, as he personally communicated with me, was reviewer of my PhD thesis (Ref 1 below), and*

*attended conferences where papers were presented (eg. ref 2).*

*Given this, the authors should thoroughly re-study existing literature and reference some key papers out of the oil and gas industry. They also should make very clear that this is not a novel/original approach but simply a standard application of what has done before and is routine in oil and gas reservoir modelling. The authors should also make clear reference to similar work already conducted ~25 years ago at the same site (Columbus Air Force Base test site where the MADE experiment was conducted), eg. ref 2.*

*The only reason why the material could be published, is that it finally may point out the scientific confusion and structural research mis-management around the MADE experiment and stochastic hydrology. The MADE experiment has led to numerous publications in journals, which all ignored to account for geological heterogeneity in an appropriate manner and ignored other work that would not fit the premises of stochastic hydrology (macro dispersion theory).*

*Ref 1 – Herweijer, J.C.,1997.Sedimentary heterogeneity and flow towards a well. Ph.D. dissertation, Free University, Amsterdam (https://www.hydrology.nl/images/docs/dutch/1997.01.07_Herweijer.pdf)*

*Ref 2 – Herweijer, J.C, 1996.Use of sedimentology and geostatistical modeling to estimate uncertainty of groundwater models. Proc. International Conferenceon Calibration and Reliability in Groundwater modeling (ModelCARE96), Golden (CO,USA), September, 1996 (https://pdfs.semanticscholar.org/a5a5/25d8da8091bb59a59795262d932f0b4a6333.pdf)*

*Please also note the supplement to this comment:*
*https://www.hydrol-earth-syst-sci-discuss.net/hess-2020-30/hess-2020-30-SC1-supplement.pdf*

We want to thank Joost Herweijer for his interest in our work. We acknowledge the initiation of the scientific discussion on the subject matter of integrating aquifer heterogeneity to hydrogeological transport models as this was one of the goals of the work. As stated in the manuscripst's last paragraph, we aim to contribute to bridging the gap between the advanced research in stochastic hydrogeology and its limited use by practitioners. In this line, we agree that hierarchical deterministic/stochastic modeling permeability is used in the oil and gas industry, but hardly found it's way into applied hydrogeology. We acknowledge his effort in providing us publications on MADE which have not been available to us or of which we were not aware, respectively.

The points he raised are now addressed in the manuscript:
- it was specified that the presented work is an application of the hierarchical approach. Reformulated e.g. as "novel conceptualization strategy of aquifer heterogeneity in a hierarchical deterministic/stochastic framework"
- missing references are integrated: Herweijer, 1996, 1997, Bryant & Flint, 2009, Bianchi & Zheng, 2016
- We are aware that this study is not the first approach to model transport at the MADE site. However, it is one of the few predictive models (no calibration) for the MADE 1 experiment and conceptually very different from the other predictive approaches [e.g. Fiori et al. 2013, 2017, Bianchi & Zheng, 2016]. Where the Monte Carlo procedure is related to computational effort, the amount of required field data is limited making the approach attractive to less investigated sites.

- The mentioned work on "A hierarchical deterministic/stochastic approach applied to tracer tests at Columbus Air Force Base" was integrated and provides a valuable reference. The application of hierarchical approaches in the context of pumping test interpretation is perfectly in line with the suggestions in our study to make use of distinct interpretation methods for aquifer heterogeneity characterization. It is a great example to make use of pumping tests to determine connected areas of high conductivity. The application to the MADE experiment however relates to another tracer test setup, with forced flow between wells, again focusing on fast flow channels.

*()*

*2D vs 3D has been looked extensively, see e.g.*

- *Static characterizations of reservoirs: refining the concepts of connectivity and continuity Joseph M. Hovadik and David K. Larue Petroleum Geoscience, Vol. 13 2007, pp.195–211 http://citeseerx.ist.psu.edu/viewdoc/download?doi=10.1.1.818.7201&rep=rep1&type=pdf*

- *King, P. R., 1990, The connectivity and conductivity of overlapping sand bodies. In, Buller Anthony T. et al.,eds, North Sea oil and gas reservoirs; II, Proceedings of theNorth Sea oil and gas reservoirs conference. [Book, Conference Document], Pages 353-362. https://link.springer.com/chapter/10.1007/978-94-009-0791-1_30*

- *Models in study @ MADE site mentioned in my previous comments were all run in 3D, as is was concluded that 2D models tend to 'suppress' connectivity.*

The discussion on the model dimensionality (2D vs. 3D) is given along the response to referee #2.

[revised manuscript text omitted]

---

## Referee Report (RR1)

**Review of:**

Zech, A, P Dietrich, S Atttinger, and G Teutsch, 2020, A Field Evidence Model: How to Predict Transport in a Heterogeneous Aquifers at Low Investigation Level?  Hydrology and Earth Sciences (in review)

By:     Dr Joost C. Herweijer – joost.herweijer@gmail.com

Date:   2 September 2020.

**Summary**

Below are the three main issues that I identified with respect to this paper:

1- The conceptualization is poor, it does not take into account available geological information. The modelling process does not adhere to well-published standard practices for this type of hierarchical modelling. Some key features of the data which underpin the conceptualization are not addressed, most critically, inconsistencies regarding the basic data (K values).
2- The analysis is conducted in a single 2D-vertical cross-section assuming symmetry in one of the horizontal directions. This is not in accordance to observed piezometric level and tracer test data showing flow and transport in both horizontal directions. The results of the 2D model cannot be used for quantitative analysis.
3- The paper incompletely references geological issues very relevant to conceptualization of heterogeneous aquifers.  Especially when it comes to options for geological analysis and modelling, some statements in the paper are not well documented and potentially misleading

Based on point 1 and point 2, I reject this manuscript.

The MADE data set is the result of significant efforts to collect a 3D dataset of hydraulic conductivity and tracer transport observations. Many papers have been published addressing various aspects of the data and the geological concepts, showing significant variability in 3D of hydraulic conductivity and tracer transport. Creating models in 3D that take into account these data is well within the realm of what is technically possible. No scientific rationale is presented to explain why the model has been restricted to only 2D.

**1 – Conceptualization is incomplete, inadequate and not novel**

The paper promises a 'novel' conceptualization. However, the paper falls short of an adequate conceptualization. As discussed in 1a below, very adequate conceptualizations have been provided for the MADE aquifer (eg. Herweijer, 1997) and the MADE site in particular (eg. Julian et al., 2001), and these should be referenced and compared with the proposed approach. Also, as discussed in 1d below, the 'nested' scale' approach chosen by the authors is not novel.

1a) Creating K zones based on the change in piezometric surface was earlier discussed for the MADE site by Rehfeldt (1992) and Herweijer (1997). These K zones are related to the main geological feature at the site that has been reported about in several papers (eg. Herweijer and Young, 1991; Young 1995; Herweijer, 1996, 1997; Julian 2001; Bowling et al., 2005). The single vertical K zone model presented in the paper is incorrect. Herweijer (1997) established that the MADE aquifer consists of a two layer system where a high K channel deposit incised in somewhat older lower K deposits (see figure below).

[Figure]

[Figure]

Aerial photograph of Columbus Air Force Base and the MADE and the MADE-1HA test site.

Also visible is the paleo-channel that intersects both test sites.

Reproduced from Herweijer, 1996 (fig 1&2) and 1997 (fig 6.25) showing a 'hierarchical approach combining large-scale deterministic structures and simple stochastic methods'. This model was developed for the MADE-1HA site 100 m away from the MADE site

Bowling et al. (2005) shows via GPR data a similar two-layer system with a unit of coarser sand and high energy depositional bed-forms overlaying a more stratified lower sand unit. The same two-layer system can also be seen in figure 2 of the paper under review, as the borehole flowmeter K data of wells F20 and F40 show a sharp increase of K above the 56-67 m depth level.

The two-layer system is also clearly reflected in the Tritium plume of the MADE-2 experiments (see figure below from Boggs et al., 1993).

[Figure]

*In order to obtain a much more realistic conceptual model, the authors should use the above references on the geological background of the high K contrast. The conceptual model should also include the vertical K contrast at ~ the 57-58 m depth level as indicated by figure 2 in their paper and shown by various references cited above. The paper should also present a cross-section of the modelled vertical tracer distribution to compare with the observed vertical tracer distribution.*

1b)  At the scale below the main zonation scale referred to in 1a above, the authors then insert binary K contrast 'inclusions' representing a medium scale of heterogeneity. This binary (Boolean) technique has been commonly used (eg. Haldorsen and Lake, 1984, Desbarats 1987) to create a heterogeneous architecture at various scales. The dimensions of these 'inclusions' assumed by the authors, seem not to be based on any field evidence or analogue systems. Rehfeldt (1992) shows a section of a nearby quarry with potential dimensions of these inclusions. Herweijer and Young (1991) show how pumping test data reveal some insights regarding the hydraulic continuity of these inclusions. Herweijer (1997) specifically mentions some scenarios for dimensions based on sedimentary analogues for the same aquifer.  Bowling (2005) shows detailed data for the same site, where GPR sections show some of the sedimentary structures controlling heterogeneity on this scale

*In order to improve the conceptual model, the authors should elaborate on the nature of these inclusions and use the dimensional information for the inclusions contained in the above references.*

1c) At the next lower scale level, the authors use randomized K values from the borehole flow meter data.  Figure 4 shows several datasets for K distribution, and there is a significant discrepancy between the mean and the range of borehole flowmeter data and the K datasets. For the borehole flowmeter the log-normal mean is a factor 5 lower than the pumping test value that represents the bulk of these BHF data (the pumping test in the high K zone – the channel deposit). The mean of the K derived from grain-size  is a factor 2 higher than the pumping test value, which could indicate that the K values derived from grainsize would be more reliable to represent the high end of K values. This type of differences between hydraulic conductivity data is not uncommon. Apart from data acquisition issues, the differences between differently measured K values can often be traced back to scale effects, which are of utmost relevance to conceptualization, and should be addressed in the paper. The authors should also review for example Rehfeldt ea. (1989) and Young (1998) regarding some issues with the borehole flowmeter data specific to the MADE aquifer. Young (1995) and Herweijer (1997) show at the neighbouring MADE-1HA test site borehole flowmeter K-values for the same sediments. They publish values with a higher log-normal mean and maximum, which corroborate the grain-size K values for the MADE site as shown in figure 4 of the paper under review. This extreme end of the K distribution has at the MADE-1HA site a significant effect forming high-K pathways (Young, 1995; Herweijer, 1997)

*In order to support the conceptual model, the authors should review the meaning of the different hydraulic conductivity values as measured for the MADE site and how that impacts the conceptual model. They also should explain specifically why the borehole flowmeter data were selected, why the deviation between the mean of the borehole flowmeter data and the relevant pumping test is acceptable and, why the borehole flowmeter data are preferred to the grainsize K data (which seem to better represent this pumping test and the high end K values).*

1d) The hierarchical/nested scale approach using deterministic zonation with various levels of binary and continuous stochastic infill is not 'novel'. It  has been widely used before, and the authors should reference some earlier work applying this approach (see eg: Damsleth et al, 1990; Herweijer, 1997 section 6.6, specific to the MADE aquifer – see also first figure of this review; Smith et al., 2001; Yupeng & Shenhe, 2013)

*The authors should quote above references as examples of the hierarchical method they employ, and not refer to their approach as novel, neither in general nor specific to the MADE aquifer.*

**2 – Use of 2D model for a 3D plume**

The paper presents a 2D cross-sectional model along the main axis of flow.

The field data show a major component of flow transversal to the main flow. The tritium plume picture below from Boggs et al., 1993) shows that initially the tracer released at the 5 injection wells converges and subsequently diverges. An elongated finger of the plume shows further downstream, but probably already developed closer to the source area. This finger is probably related to some very high K pathways related to sedimentary structures that are highly anisotropic and directionally variable (Young, 1995; Herweijer, 1997).

[Figure]

Figure 4-2  Horizontal Sections Through Tritium Plume at Elevation 59.5 m at 27, 132, 224, and 328 Days

X

The Bromide plume shows similar transversal movement (including a sharp sideways movement close to the source) and downstream patchiness (see eg. Julian et al., 2001, fig 5 &13)

[Figure]

Figure 5.  Depth-averaged total BTEX (left) and bromide (right) concentration distributions 278 days after source emplacement.

[Figure]

Figure 13.  Comparison of observed and calculated bromide peak concentrations for (a) snapshot 3, 152 days after source release, and (b) snapshot 4, 278 days after source release.

The piezometric surface map around the injection zone (Rehfeldt et al., 1992, see figure below) shows a complex convergent and then divergent pattern. This is probably due to refraction of the

flow lines at the boundary of two zones with very different K values and which occurs at an angle to the regional flow direction (Freeze and Cheery, 1979).

[Figure]

*Given a clear horizontally anisotropic flow and transport pattern, a 2D analysis/model is very limited and unrepresentative. A 3D model should be used. The results of the 2D model cannot be used for quantitative analysis.*

**3 – Potentially misleading remarks re use of geological data and geological models**

The paper makes (line 53-57, 64) statements re. the use of geological data (training images) and geological models referring papers that are 20+ years old (Koltermann and Gorelick, 1996; Herweijer, 1997).

In line 54, the authors dismiss training images as limited available and unrepresentative. This statement is incorrect: training images are quite widely available sourced from satellite images (Google Earth) and extensive literature on geology, sedimentology and paleogeography. This holds especially for relative recent shallow deposits which form the MADE aquifer, and which are often the subject of groundwater modelling efforts.

Herweijer (1997) provides a number of references to the sedimentology and paleogeography of the MADE site, which would provide a very good start with respect to representative training images. Ronayne et al. (2008) give a good example of a model based on training images to model a hydrogeological test site.

The paper also states that geological models as used in the petroleum industry have not found their way into applied hydrogeology (line 64), a statement which is quite strong, and in my view incorrect. Alloisio, (2011), Dowling et al. (2013) and Peereboom (2018) are examples of applications for a variety of shallow to deep aquifers. Even if it is the case that this type of modelling has not 'found its way' into widespread use in hydrogeology, this is not a reason to simply to set it aside from a research point of view. The authors should explain why their approach is 'better' than the standard geological modelling methods used in the petroleum industry and the hydrogeological applications of these methods referenced earlier in this paragraph.

*The authors should re-assess the literature on the above matter and correct their statements about the use of geological data and models.*

**Some further issues in need of clarification**

**Line 335 and last section of supplement**

As earlier discussed, the MADE plumes are variable in 3D and should really be modelled in 3D

The sections explaining the 3D modelling effort to confirm the validity of the 2D model are confusing. It is unclear if the 2D model was simply copied in the 3$^{rd}$ dimension, ie. is completely symmetric in the 3$^{rd}$ dimension, or if some sort of heterogeneity in the 3$^{rd}$ dimension was included. The supplement also states that the extension in the 3$^{rd}$ dimensions has no impact because of the binary nature of the 'inclusions' vs. K values that change gradually. If transport being restricted to a 2D cross-section of a 3D model is a significant impact of the binary conceptualization, than the binary conceptualization should not be used, as gradual changes of K are the norm for non-fractured/fissured aquifers such as the MADE aquifer

It would also be informative if the authors would present a visualization (map or x-section slices) of the 3D model and the modelled plume. It should also be looked at if a small level of transversal dispersion (representing very small scale heterogeneity) would have a significant impact on the 3D version of the 2D model.

**Line 330- 334 and section 'details on flow and transport' section in supplement**

The injection rate modelled is Qin = 1.166e-5 m3/sec. The injection rate quoted by Boggs et. al. is 10.07 m3/48.5 hr which converts to 5.57e-5 m3/sec. If any adjustment has been made to the injection rate (perhaps to adapt for the 2D model setting?) this has to be clarified.

The paper states: *We use a flux related injection representing natural conditions. For technical details, the reader is referred to the Supporting Information.* This is unclear and there seems to be no further specific discussion on '*flux related injection'* in the supporting documentation.

**Section 3.4 - calibration and predictive capacity of model**

The model matches data in a line downstream of the injection, but as discussed earlier in the review (point 2) does not represent any 3D movement of the plume close to the injection point and further downstream. The paper also does not show a vertical cross-section of the observed vs the modelled tracer distribution. Hence any calibration is quite limited and potentially has limited predictive value

As discussed in the paper (line 270) there is significant uncertainty due to the mass balance issues with the bromide tracer (~50% in snapshots past 300 days). The calibration could be (should be?) tested using the MADE-2 tritium tracer test (Boggs et al., 1993), which seems to have resulted in a better mass balance (77% in final snapshot at 328 days).

**Title**

There is a grammar error in the title (the part which reads 'in a Heterogeneous Aquifers'). It should be either 'in a Heterogeneous Aquifer' or 'in Heterogeneous Aquifers'

**References**

Alloiso, S, 2011, Integrated use of Petrel© and Modflow in the modelling of water injection and effects on a Quaternary aquifer - https://www.researchgate.net/publication/263611730_Integrated_use_of_PetrelC_and_Modflow_in_the_modelling_of_water_injection_and_effects_on_a_Quaternary_aquifer_Usage_integre_de_PetrelC_et_Modflow_dans_la_modelisation_de_l'injection_d'eau_et_effets_dan

Boggs, J.M, Beard, L.M., Waldrop, W.R., 1993 Transport of tritium and four organic compounds during a natural-gradient experiment (MADE-2), report TR-101998, EPRI, Palo Alto, CA, USA.

Bohling, G. C., Liu, G., Knobbe, S. J., Reboulet, E. C., Hyndman, D. W., Dietrich, P., & Butler, J. J. (2012). Geostatistical analysis of centimeter-scale hydraulic conductivity variations at the MADE site. Water Resources Research, 48(2). doi:10.1029/2011wr010791

Bowling, J. C., Rodriguez, A. B., Harry, D. L., & Zheng, C. (2005). Delineating Alluvial Aquifer Heterogeneity Using Resistivity and GPR Data. Ground Water, doi:10.1111/j.1745-6584.2005.00103.x

Damsleth, E.; Tjolsen, C. B.; and Omre, K. H., 1990, A two stage stochastic reservoir model applied to a North sea reservoir: SPE paper 20605 presented at the 1990 SPE Annual Technological Conference and Exhibition, New Orleans, La., September 23-26.

Desbarats, A.J., 1987, Numerical estimation of effective permeability in sand-shale formations: Water Resources Research, v. 23, no. 2, p. 273-286.

Desbarats, A.J., 1990, Macro-dispersion in sand-shale sequences: Water Resources Research, v. 26, no. 1, p. 153-163.

Dowling, J. , B. Zimmerlund, J. Keneally, E. van den Burg, 2013. An integrated model for mine dewatering at the Bagdad mine. In Wolkensdorfer, Brown and Figuora (editors), INWA, https://www.imwa.info/docs/imwa_2013/IMWA2013_Dowling_545.pdf

Freeze R.A. and Cherry J.A., 1979, Groundwater, Prentice Hall

Haldorsen, H. H., & Lake, L. W. (1984). A New Approach to Shale Management in Field-Scale Models. Society of Petroleum Engineers Journal, 24(04), 447–457. doi:10.2118/10976-pa

Herweijer, J.C., and Young, S.C., 1991, Use of detailed sedimentological information for the assessment of aquifer tests and tracer tests in a shallow fluvial aquifer: Proceedings of the 5th Annual Canadian American Conference on Hydrogeology, Calgary, September, 18-20, 1990. NWWA, Dublin (Ohio)

Herweijer, J.C, 1996.  Use of sedimentology and geostatistical modelling to estimate uncertainty of groundwater models. Proc. International Conference on Calibration and Reliability in Groundwater modelling (ModelCARE 96), Golden (CO, USA), September, 1996. https://pdfs.semanticscholar.org/a5a5/25d8da8091bb59a59795262d932f0b4a6333.pdf

Herweijer, J.C., 1997. Sedimentary heterogeneity and flow towards a well. Ph.D. dissertation (in English), Free University, Amsterdam, NL, https://www.hydrology.nl/images/docs/dutch/1997.01.07_Herweijer.pdf

Julian, H. E., Boggs, J. M., Zheng, C., & Feehley, C. E. (2001). Numerical Simulation of a Natural Gradient Tracer Experiment for the Natural Attenuation Study: Flow and Physical Transport. Ground Water, 39(4), 534–545. doi:10.1111/j.1745-6584.2001.tb02342.x

Koltermann, C. E. and Gorelick, S. M.: Heterogeneity in Sedimentary Deposits: A Review of Structure-Imitating, Process-Imitating, and Descriptive Approaches, Water Resour. Res., 32, 2617–2658, https://doi.org/10.1029/96WR00025, 1996.

Peereboom E, 2018, Exploring electrofacies for property mapping of the Triassic: an improved approach for geothermal development in Brabant, MSc Thesis (in English),  University of Utrecht, NL

Rehfeldt KR, Hufschmied P, Gelhar LW, Schaefer ME, 1989, Measuring hydraulic conductivity with the borehole flowmeter, Report EN-6511, EPRI, Palo Alto, CA, USA.

Rehfeldt, K.R.; Boggs, J.M.; and Gelhar, L.W., 1992, Field study of dispersion in a heterogeneous aquifer – 3 - Geostatistical analysis of hydraulic conductivity: Water Resources Research, v. 28, no. 12, p 3309-3324.

Ronayne, M.J., S.M. Gorelick, and J. Caers. 2008. Identifying discrete geologic structures that produce anomalous hydraulic response: an inverse modeling approach. Water Resources Research 44. DOI:10.1029/2007WR006635

Smith, R., W.A. Bard, J. Corredor, J.C. Herweijer, S.R. McGuire, A. Antunez, T. Block, N. Lazarde, 2001. Geostatistical Modeling and Simulation of a Compartmentalized Deltaic Sequence, Ceuta Tomoporo Field, Lake Maracaibo, Venezuela. SPE paper 69572, presented at the 2001 SPE Latin American and Caribbean Petroleum Engineering Conference, Buenos Aires, Argentina, 25–28 March 2001

Young, S.C., 1995, Characterization of high-K pathways by borehole flowmeter and tracer tests: GroundWater, v. 33, no. 2, p. 311-318
https://ngwa.onlinelibrary.wiley.com/doi/abs/10.1111/j.1745-6584.1995.tb00286.x

Young, S.C., 1998, Impacts of Positive Skin Effects on Borehole Flowmeter Tests in a Heterogeneous Granular Aquifer GroundWater, v. 36, no. 1, p. 67- 75
https://ngwa.onlinelibrary.wiley.com/doi/epdf/10.1111/j.1745-6584.1998.tb01066.x

---

## Author Response (AR2)

**Rebuttal**

Dear editor and referees,

thank you for the time and effort you put into reviewing our manuscript. In the following, we provide point-by-point replies (in blue) to the comments (in italic). While line numbers in the comments refer to the previous manuscript version, mentioned lines in the response relate to the revised manuscript. Attached to the response is a marked-up manuscript version with tracked changes.

With kind regards,

Alraune Zech
on behalf of the author-team.

**Editor comment:**

*Dear Authors:*
*Your revised documents, together with your rebuttal, were evaluated by four reviewers, three of them had already participated in the discussion step of the journal. Three out of the present reviewers evaluated fairly well the scientific quality of your revised paper and the presentation of the relevant results. Instead, the scientific significance of your study received contrasting comments, and one reviewer was quite critical with respect to both the scientific significance and quality of your study.*

*After having read again all the documents pertaining to your submission, I recognize that you put a great effort into improving the original paper, by providing also satisfactory responses to all of the comments and concerns raised during the discussion step. However, I also concur with the present reviewers that even this revised version is still not in a shape to be published in HESS and should require some additional revisions.*

*By sharing the comment from one reviewer, I do believe that the present contribution is definitely not the only example of a hierarchical approach to transport in heterogeneous porous materials and soils, and it will not be an isolated case, I guess (to the benefit of Science). Anyhow, I do suggest the Authors should carefully consider revising their paper according to the concerns and points now raised by Ref's. #2 and #3. Moreover, due attention should be given to all of the comments received by Ref. #4.*

*Therefore, you are invited to upload new revised documents, together with point-by-point replies to all of the comments received by the reviewers. Should you disagree with some comments, please explain why clearly.*

We followed the advice of the editor and revised the manuscript accordingly to the comments raised by the referees, particularly those of Ref. #3. While we adapted the manuscript to the criticism raised by Ref. #4, we see a significant discrepancy in the referee's and our perception of the work, including its purpose and scientific methods. We provide detailed responses to all points raised by Ref. #4 within this rebuttal.

**Referee #1 Evaluation:** *For final publication, the manuscript should be accepted as is.*

**Referee #2 (X. Sanchez-Vila) Evaluation:** *For final publication, the manuscript should be accepted as is.*

**Referee #3 Evaluation:** *For final publication, the manuscript should be accepted subject to **minor revisions.***

*This is an interesting paper trying to prove that stochastic groundwater modelling is possible. The authors discuss a modular approach to incorporate heterogeneity into groundwater flow modelling. Then they apply to one of the experiments in the well-known MADE site. I recommend publication after some "minor" corrections, in the sense that they will take little time to implement, but "major," in the sense that they lower the author's claims. I like the paper because it shows that stochastic modelling is possible. I do not like some of the claims and discussions. I do not like that the model is two-dimensional, either.*

We want to thank the referee for his positive evaluation of our work. We appreciate the time and effort he put into reviewing our manuscript. The paper will benefit from revising it according to his constructive comments. We toned down the claim on originality with respect to hierarchical aquifer modelling and specified the purpose of the study alongside: we aim to provide an easy-applicable conceptualization for integrating heterogeneity quantitatively into models in line with the referee's statement "*that stochastic modelling is possible*". To underline that, we also made the numerical code for generating random binary inclusion structures public with reference provided in code availability section.

*MAJOR CORRECTIONS*

1) *There is nothing new in this modular approach to incorporating heterogeneity into aquifer modelling. Any claim of originality in this respect should be toned down, and references to similar approaches in groundwater modelling or reservoir engineering included. (A few references implementing this concept could be Damsleth et al., 1992, Huysmans and Dassargues, 2009, Neto et al., 1994, Proce et al., 2004, to cite a few dating back to the past century). Please, rewrite lines 64-65 with proper referencing.*

   We reformulated the abstract and other parts of the manuscript (particularly regarding the use of "novel") and specified the advantage of the modular approach we present.

   We reformulated (previous) lines 64-65 and integrated (provided and additional) references.

2) *Already in the abstract, the authors claim that their model is constructed with as minimal data as possible. This is clearly not so in the description of the application. The amount of data used is substantial and hardly available in most sites. Even if some of the data are not used as conditioning data, they are needed to infer the different parameters of the nested heterogeneous models.*

   In the course of the manuscript, we introduce several conceptual models for heterogeneity structure, with different levels of observation data requirements:

   1) deterministic (module A): piezometric surface map, pumping tests

   2) deterministic + binary (modules A+B): piezometric surface map, pumping tests, few flowmeter logs

3) deterministic + binary + random log-normal (modules A+B +C): piezometric surface map, pumping tests, multiple flowmeter logs (for geostatistical analysis)

While we agree with the referee on the aspect of available data for the third conceptual model (modules A+B +C), we think that the first two are based on a decent amount of field data which is often available at field site. Head observations and a few pumping tests are rather standard. While flowmeter logs might not always be available, there is often some geological information on contrasting layering of sand and clay. Nowadays, few depth profiles are easy to achieve from direct-push injection logging (DPIL)/hydraulic profiling (HPT) or cone-penetration tests (CPT).

However, we see that this was not outlined properly. For clarification, we specified that the level of data requirement is indeed high for the concepts including module C. Also, we agree that the word "minimal" is misleading in this context. It was thus eliminated in the text and sentences were rephrased. E.g. we reformulated the abstract: "The conductivity model is constructed step-wise following field evidence from observations; seeking a balance between model complexity and available field data."

3) *In the introduction, some statements should be toned down, and a few historical references are missing.*

The corresponding part of the introduction was revised considerably.

1) *In line 40, it says that huge amounts of data are needed for kriging. Certainly, you need data to infer the variograms, but not as many as the one you need in the latter application, where you claim that hardly any data are used.*

The passages on Kriging were reformulated. We specified the aspect of required data alongside those of Gaussian models. See also the previous comment.

2) *In line 42, it says that stochastic methods, on the other hand, need a limited amount of data. Kriging is a stochastic method, which apparently needs a huge amount of data. In any case, the statement is not true, stochastic methods need data, large amounts, as it is shown later.*

We rephrased this inconsistent paragraphs profoundly.

3) *When listing the common methods, some historical references are missing, such as Gómez-Hernández and Gorelick, 1989, for Gaussian random fields; Journel and Gómez-Hernández, 1990, for indicator simulation; and Strebelle for multiple-point statistics. (By the way, Freeze, 1975 is not the best example of the use of Gaussian random fields, since he used uncorrelated values.)*

We rephrased and modified the references accordingly.

4) *Line 76, it seems that the number of data used is not minimal. Previous works have shown that properly accounting for hydraulic conductivity heterogeneity at the MADE site is sufficient to reproduce the mass transport behaviour at the MADE site (i.e., Salamon et al. (2006), or Li et al. (2011))*

The paragraph was rephrased, missing references were added.

4) *For the paper to really serve its purpose of enticing practitioners to use stochastic modelling, the model would have had to be three-dimensional. But it is not! How much does the dimensionality reduction influence the results? This must be discussed.*

We extended the discussion on the impact of dimensionality reduction and clarified the difference between heterogeneity conceptualizations dominated by the binary structure and a log-normal distribution: When it comes to a predominantly log-normal heterogeneity structure, we agree that dimensionality makes a difference. When module C is the main component representing heterogeneity, models should actually be in 3D to not underestimate flow velocity and connectivity. For conductivity conceptualizations dominated by the binary structure (module B), the differences between model results for 2D and 3D are marginal. As the case for your application to the MADE site. This is the results of the binary layer structure, which does not increase connectivity in the third (y-)dimension when considering horizontal isotropy. In this sense, the 2D character of binary fields can even be more enticing for practitioners to use stochastic modelling at this reduced computational effort. However, we stressed, that when applying the proposed heterogeneity conceptualization for modelling flow in transport in other application, a 3D model setup should be considered first and a complexity reduction to 2D models should only be taken when warranted by the conductivity conceptualizations.

We revised the paragraph in section 3.3 Numerical Model Setting, adapted the supporting information and added a paragraph in the discussion section 4.

**References** (provided by the referee)

Damsleth, E., Tjolsen, C. B., Omre, H., & Haldorsen, H. H. (1992). A two-stage stochastic model applied to a North Sea reservoir. Journal of Petroleum Technology, 44(04), 402-486.

Gómez-Hernández, J. J., & Gorelick, S. M. (1989). Effective groundwater model parameter values: Influence of spatial variability of hydraulic conductivity, leakance, and recharge. Water Resources Research, 25(3), 405-419.

Li, L., Zhou, H., & Gómez-Hernández, J. J. (2011). A comparative study of three-dimensional hydraulic conductivity upscaling at the macro-dispersion experiment (MADE) site, Columbus Air Force Base, Mississippi (USA). Journal of Hydrology, 404(3-4), 278-293.

Huysmans, M., & Dassargues, A. (2009). Application of multiple-point geostatistics on modelling groundwater flow and transport in a cross-bedded aquifer (Belgium). Hydrogeology Journal, 17(8), 1901.

Journel, A. G., & Gomez-Hernandez, J. J. (1993). Stochastic imaging of the Wilmington clastic sequence. SPE formation Evaluation, 8(01), 33-40.

Neton, M. J., Dorsch, J., Olson, C. D., & Young, S. C. (1994). Architecture and directional scales of heterogeneity in alluvial-fan aquifers. Journal of Sedimentary Research, 64(2b), 245-257.

Proce, C. J., Ritzi, R. W., Dominic, D. F., & Dai, Z. (2004). Modelling multi-scale heterogeneity and aquifer interconnectivity. Groundwater, 42(5), 658-670.

Salamon, P., Fernandez-Garcia, D., & Gómez-Hernández, J. J. (2007). Modelling tracer transport at the MADE site: the importance of heterogeneity. Water resources research, 43(8).

Strebelle, S. (2002). Conditional simulation of complex geological structures using multiple-point statistics. Mathematical geology, 34(1), 1-21.

**Referee #4 (J. Herweijer) Evaluation:** *For final publication, the manuscript should be **rejected.***

Summary

*Below are the three main issues that I identified with respect to this paper:*

*1- The conceptualization is poor, it does not take into account available geological information. The modelling process does not adhere to well-published standard practices for this type of hierarchical modelling. Some key features of the data which underpin the conceptualization are not addressed, most critically, inconsistencies regarding the basic data (K values).*
*2- The analysis is conducted in a single 2D-vertical cross-section assuming symmetry in one of the horizontal directions. This is not in accordance to observed piezometric level and tracer test data showing flow and transport in both horizontal directions. The results of the 2D model cannot be used for quantitative analysis.*
*3- The paper incompletely references geological issues very relevant to conceptualization of heterogeneous aquifers. Especially when it comes to options for geological analysis and modelling, some statements in the paper are not well documented and potentially misleading.*

*Based on point 1 and point 2, I reject this manuscript.*

*The MADE data set is the result of significant efforts to collect a 3D dataset of hydraulic conductivity and tracer transport observations. Many papers have been published addressing various aspects of the data and the geological concepts, showing significant variability in 3D of hydraulic conductivity and tracer transport. Creating models in 3D that take into account these data is well within the realm of what is technically possible. No scientific rationale is presented to explain why the model has been restricted to only 2D.*

Given the preceding discussion and the harsh critique of the referee, we are afraid that nothing we modify within the manuscript will lead to a different general evaluation of the referee. However, we tried our best to address all points raised and adapt the manuscript.

First, we like to address two key points, which we feel are the major source for misunderstandings and disagreement between the referee and us:

- **purpose of the manuscript:** we aim to provide an easy-applicable conceptualization for integrating heterogeneity quantitatively into flow and transport models which naturally differs from previously presented hierarchical approaches. We did **not aim** to provide a detailed heterogeneity conceptualization for MADE, e.g. **reconstruction of 3D structure**, which produces the actual 3D tracer plume of the transport experiment at that side. We follow a different path of heterogeneity conceptualization by generating **random conductivity** structures (in combination with Monte Carlo simulations) which capture the main feature of transport observed at the site which a decent amount of field data. By construction, these structures do not necessarily represent the actual aquifer structure at MADE.

- **Data availability in (hydrogeological) field sites** is well known to be much more reduced (given budget limitation) than in petroleum industry related research. In this line, we **purposely** did **not** use all (hydro-)geological data available for MADE. We rather tried to rely on a decent amount of data which is accessible through standard and/or (cost-efficient) novel monitoring methods at usual hydrogeological sites.

We see that these points might not have been clearly enough addressed in the manuscript. We thus, emphasized them in the revised manuscript version.

**Detailed response:**

1 – Conceptualization is incomplete, inadequate and not novel

We want to stress the point, that a model is always a simplification of the reality. We address the question: How simple can it be to still represent the relevant processes and to make reasonable predictions (given the aim of the study)? As addressed in the manuscript in l 256: *"We thereby aim to identify the "most simple" of our concepts which still provides a reasonable prediction of the complex observed mass distribution."* We will refer to the aim of a "most simple" model conceptualization repeatedly.

We agree with the referee that (at an arbitrary site) a detailed site conceptualization is best when considering all available (geological) data. We are aware that several detailed conceptualizations of the heterogeneous structure for the MADE site exist including the work of the referee (Herweijer, 1997) or Dogan et al., (2011, 2014). However, a detailed model of the MADE site was not the purpose of our work. On the contrary, we aimed to present a simple approach for flow and transport modelling taking heterogeneity into account to reproduce non-uniform transport plumes with a decent level of field observation data. We chose the MADE site since it provides ample references data and studies to compare to.

This is explicitly stated in the manuscript several times: l.9, l. 90ff, l. 305ff (to just state a few).

*The paper promises a 'novel' conceptualization. However, the paper falls short of an adequate conceptualization. As discussed in 1a below, very adequate conceptualizations have been provided for the MADE aquifer (e.g. Herweijer, 1997) and the MADE site in particular (e.g. Julian et al., 2001), and these should be referenced and compared with the proposed approach. Also, as discussed in 1d below, the 'nested' scale' approach chosen by the authors is not novel.*

We rectified the formulation "novel". We are aware that the conceptualization of heterogeneity into zones and binary structures is conceptually not new. Also its application to MADE is known to us. The novel aspect in our approach is the **quantitative** use of these components for a **predictive** model (without calibration) and giving guidance to use this conceptualization for transport models also at other sites, including software tools.

A comparison of your approach to the detailed aquifer conceptualizations, such as presented by Herweijer, 1997 is neither meaningful nor feasible:

- representation of aquifer heterogeneity differs conceptually: while Herweijer, 1997 aims to provide a reconstruction of the actual heterogeneity pattern, we do not, but make use of ensembles of random structures which do not directly represent the actual structures.

- quantitative transport model: our target is to quantitatively model the transport in a predictive manner (without calibration), while Herweijer, 1997 does not provide a quantitative transport model: *"It should be noted that the data are only used in a qualitative manner; in other words, no analysis method is attempted that aims at a direct quantitative duplication of field data".,* p. 98 of the thesis.

Similarly, the work of Julian et al., 2001 does not target at predictive modelling either but on structure reconstruction: *"Inverse analysis was conducted to estimate optimal K values"*, p. 543.

However, we provide direct reference to both publications.

*1a) Creating K zones based on the change in piezometric surface was earlier discussed for the MADE site by Rehfeldt (1992) and Herweijer (1997). These K zones are related to the main geological feature at the site that has been reported about in several papers (e.g. Herweijer and Young, 1991; Young 1995; Herweijer, 1996, 1997; Julian 2001; Bowling et al., 2005). The single vertical K zone model presented in the paper is incorrect. Herweijer (1997) established that the MADE aquifer consists of a two layer system where a high K channel deposit incised in somewhat older lower K deposits (see figure below).*

*Bowling et al. (2005) shows via GPR data a similar two layer system with a unit of coarser sand and high energy depositional bed-forms overlaying a more stratified lower sand unit. The same two-layer system can also be seen in figure 2 of the paper under review, as the borehole flowmeter K data of wells F20 and F40 show a sharp increase of K above the 56-67 m depth level.*

*The two-layer system is also clearly reflected in the Tritium plume of the MADE2 experiments (see figure below from Boggs et al., 1993).*

***In order to obtain a much more realistic conceptual model, the authors should use the above references on the geological background of the high K contrast. The conceptual model should also include the vertical K contrast at ~ the 57   58 m depth level as indicated by figure 2 in their paper and shown by various references cited above. The paper should also present a cross-section of the modelled vertical tracer distribution to compare with the observed vertical tracer distribution.***

The two zone model is a simplification of the complex geological structure at the MADE site which represents the large scale zonation indicated by head observations to conductivity conceptualizations,  but of course not the actual complex pattern. We added the reference to Rehfeldt et al., 1992 explicitly for zonation based on changes in the piezometric surface.

Again, we refrain from using (and referring) to detailed investigations on geological structure for the conceptual conductivity setup to remain exemplarily for concept use at other (less intensively investigated) sites. We do not aim to construct a more realistic conceptual model, particularly not by calibrating to transport experiment results. We seek the "most simple" concept which still provides a reasonable prediction of the complex observed mass distribution. A comparison of modelled plume cross-sections does not make sense to us. At the level of only Module A it is a deterministic Gaussian plume (following the analytical solutions of ADE). For the conductivity conceptualizations including random components (Module B and/or C), plume simulation results from individual realizations do not reproduce the observed tracer distribution, as a result of the random conceptualization. We further refrain from including comparison to cross-section observations since we do not aim to reproduce the actual tracer plume, but focus on predicting the average longitudinal mass distribution.

*1b) At the scale below the main zonation scale referred to in 1a above, the authors then insert binary K contrast 'inclusions' representing a medium scale of heterogeneity. This binary (Boolean) technique has been commonly used (e.g. Haldorsen and Lake, 1984, Desbarats 1987) to create a heterogeneous architecture at various scales. The dimensions of these 'inclusions' assumed by the authors, seem not to be based on any field evidence or analogue systems. Rehfeldt (1992) shows a section of a nearby quarry with potential dimensions of these inclusions. Herweijer and Young (1991) show how pumping test data reveal some insights regarding the hydraulic continuity of these*

*inclusions. Herweijer (1997) specifically mentions some scenarios for dimensions based on sedimentary analogues for the same aquifer. Bowling (2005) shows detailed data for the same site, where GPR sections show some of the sedimentary structures controlling heterogeneity on this scale.*

**In order to improve the conceptual model, the authors should elaborate on the nature of these inclusions and use the dimensional information for the inclusions contained in the above references.**

We agree that the use of binary techniques is not new. We provided reference to import statistical results, such as Rubin, 1995. We added further references in line with the suggestions by the referee.

Following our paradigm of constructing a simple structure based on a decent amount of field data, we consider the dimension of the inclusion to be unknown. Having no knowledge on horizontal correlation length or connectivity of longitudinal structures is a typical situation at sites. To cope with that we provide a strategy to come up with a reasonable range of values from those few measurements available and follow a parametric uncertainty approach: we suggest to include all of these length and determine their impact then. In this sense, the referee is right that *"The dimensions of these 'inclusions' [...] seem not to be based on any field evidence or analogue systems."* This is done on purpose and clearly outlined in the manuscript. Consequently, we do not aim to refine the conceptual model by adding information from more observational data. Again, at most field sites this amount of data (as available for MADE) is not given.
Our results further show that the precise inclusion length is not crucial for a reasonable prediction given the studies goal. The critical point is that preferential flow path are represented at all, in our case by the inclusion structure.

A clear statement on the role of field data for outlining the inclusion topology is given in the manuscript (l.191-195): *"The inclusion topology is a matter of choice and data availability. [...] More complex layering structures can be adapted if additional topological information is available. However, the specific topology often plays a subordinate role. When not having any information on spatial correlation of heterogeneity, it is beneficial to assume some instead of sticking to a homogeneous model."*

We further specified the aspect with regard to application at MADE in the manuscript (l. 303-305): *"We represent these structures making use of the binary inclusion structured described in section 2.2. We assume little to no information on horizontal structures and connectivity to mimic typical field situations – thereby deliberately ignoring the large amount of data at MADE. We make use of solely four flowmeter logs (Figure 2a)."*

*1c) At the next lower scale level, the authors use randomized K values from the borehole flow meter data. Figure 4 shows several datasets for K distribution, and there is a significant discrepancy between the mean and the range of borehole flowmeter data and the K datasets. For the borehole flowmeter the log-normal mean is a factor 5 lower than the pumping test value that represents the bulk of these BHF data (the pumping test in the high K zone – the channel deposit). The mean of the K derived from grain-size is a factor 2 higher than the pumping test value, which could indicate that the K values derived from grainsize would be more reliable to represent the high end of K values. This type of differences between hydraulic conductivity data is not uncommon. Apart from data acquisition issues, the differences between differently measured K values can often be traced*

*back to scale effects, which are of utmost relevance to conceptualization, and should be addressed in the paper. The authors should also review for example Rehfeldt ea. (1989) and Young (1998) regarding some issues with the borehole flowmeter data specific to the MADE aquifer. Young (1995) and Herweijer (1997) show at the neighbouring MADE-1HA test site borehole flowmeter K-values for the same sediments. They publish values with a higher log-normal mean and maximum, which corroborate the grain-size K values for the MADE site as shown in figure 4 of the paper under review. This extreme end of the K distribution has at the MADE-1HA site a significant effect forming high-K pathways (Young, 1995; Herweijer, 1997)*

**In order to support the conceptual model, the authors should review the meaning of the different hydraulic conductivity values as measured for the MADE site and how that impacts the conceptual model. They also should explain specifically why the borehole flowmeter data were selected, why the deviation between the mean of the borehole flowmeter data and the relevant pumping test is acceptable and, why the borehole flowmeter data are preferred to the grainsize K data (which seem to better represent this pumping test and the high end K values).**

We fully agree with the referee on the discussion on differences between hydraulic conductivity data of various monitoring methods and their relation to acquisition and scale effects. That is one reason why we included Figure 4, although not using all of the mentioned data sets. We expended the discussion on Figure 4 accordingly (l. 222ff).

The referee is mistaken in the perception that we use borehole flow meter data for the sub-scale log-normal distribution. The only parameter we deduce from field data for module C, additional to those used for module A+B, is a log-conductivity variance. Here, we refer to the most recent DPIL data of Bohling et al., 2016 (l. 343).

Furthermore, we do not use the flowmeter data for determining the mean conductivities but both pumping tests (and other data) instead (module A). We only make use of 4 flowmeter logs to deduce structural information on layering and the binary character for module B. Also note, the similarities on the two-point statistics between the different methods (Figure 4), indicating that the observation methods well agree in the structural characteristics at MADE, although differing considerably in the mean.

We clarified the relation of our model to the different mean values reported for MADE (l. 304ff): "When fixing regional conductivities from pumping tests, model scale coincides with measurement scale. This way, our structures are independent from upscaling of method (and location) specific geometric means reported for MADE (Figure 4)."

*1d) The hierarchical/nested scale approach using deterministic zonation with various levels of binary and continuous stochastic infill is not 'novel'. It has been widely used before, and the authors should reference some earlier work applying this approach (see e.g.: Damsleth et al, 1990; Herweijer, 1997 section 6.6, specific to the MADE aquifer – see also first figure of this review; Smith et al., 2001; Yupeng & Shenhe, 2013)*

**The authors should quote above references as examples of the hierarchical method they employ, and not refer to their approach as novel, neither in general nor specific to the MADE aquifer.**

We toned down the claim on originality with respect to hierarchical aquifer modelling and clarified the purpose of the study alongside: providing an easy-applicable conceptualization for integrating heterogeneity quantitatively into models.

We reformulated the abstracts and several text passages accordingly. We further integrated additional and listed references. [Note that we could not find the reference of Yupeng & Shenhe, 2013.]

2 – Use of 2D model for a 3D plume

*The paper presents a 2D cross-sectional model along the main axis of flow. The field data show a major component of flow transversal to the main flow. The tritium plume picture below from Boggs et al., 1993) shows that initially the tracer released at the 5 injection wells converges and subsequently diverges. An elongated finger of the plume shows further downstream, but probably already developed closer to the source area. This finger is probably related to some very high K pathways related to sedimentary structures that are highly anisotropic and directionally variable (Young, 1995; Herweijer, 1997).*

*The Bromide plume shows similar transversal movement (including a sharp sideways movement close to the source) and downstream patchiness (see e.g. Julian et al., 2001, fig 5 &13) flow lines at the boundary of two zones with very different K values and which occurs at an angle to the regional flow direction (Freeze and Cheery, 1979).*

***Given a clear horizontally anisotropic flow and transport pattern, a 2D analysis/model is very limited and unrepresentative. A 3D model should be used. The results of the 2D model cannot be used for quantitative analysis.***

Again, we need to outline that we do NOT consider results of transport experiments to constrain the hydraulic conductivity distribution in order to keep our model predictive and free of calibration.

The referee nicely outlined how the results of the transport experiments revealed information on preferential flow and non-uniform flow in transverse horizontally direction. BUT this could not have been foreseen in the hydraulic observation data. This is confirmed by the setup of the monitoring network for the first transport experiment which relied on hydraulic data only.

We assume symmetry in the horizontal direction because hydraulic heads and hydraulic conductivity do not indicate anisotropy in horizontal direction. The observed piezometric levels (Figure 1, left) show a slight non-uniformaty in the horizontal flow pattern, but in general there is a clear main flow direction perpendicular to the head isolines. Thus, a complexity reduction from 3D to 2D in terms of flow is warranted.

The impact of a 2D instead of 3D model with regard to transport was studies. In short, we found that a 2D model is sufficient to resolve the binary structure we propose. We extended the discussion of that aspect further in the manuscript (sections 3.3 & 4) and in the supporting information: We specified details in on the 3D tests and clarified the difference between heterogeneity conceptualizations dominated by the binary structure and a log-normal distribution: When it comes to a predominantly log-normal heterogeneity structure dimensionality makes a difference. When module C is the main component representing heterogeneity, models should actually be in 3D to not underestimate flow velocity and connectivity. For conductivity conceptualizations dominated by the binary structure (module B), the differences between model results for 2D and 3D are marginal. As it is the case for your application to the MADE site. This is the results of the binary layer structure, which does not increase connectivity in the third (y-)dimension when considering horizontal isotropy. In this sense, the 2D character of binary fields can even be more enticing for practitioners to use stochastic modelling at this reduced computational effort. However, we stressed, that when

applying the proposed heterogeneity conceptualization for modelling flow in transport in other application, a 3D model setup should be considered first and a complexity reduction to 2D models should only be taken when warranted by the conductivity conceptualizations.

Given the well fit of the predictive model results to observation data, we think that our model can in fact be used for a quantitative analysis. It stands in line with other transport models, explaining the complex flow patterns by an alternative conceptualizations and field data at MADE. At no point we wish to diminish the worth of any other model for the MADE site (including the work of the referee). We provide an alternative approach for predictive transport modelling at a significantly heterogeneous site with a simple conceptualization and decent observation effort.

3 – Potentially misleading remarks regarding use of geological data and geological models

*The paper makes (line 53-57, 64) statements regarding the use of geological data (training images) and geological models referring papers that are 20+ years old (Koltermann and Gorelick, 1996; Herweijer, 1997).*

*In line 54, the authors dismiss training images as limited available and unrepresentative. This statement is incorrect: training images are quite widely available sourced from satellite images (Google Earth) and extensive literature on geology, sedimentology and paleogeography. This holds especially for relative recent shallow deposits which form the MADE aquifer, and which are often the subject of groundwater modelling efforts.*

We reformulated these paragraph profoundly, also according to the comments of the referee. We want to stress that we do not dismiss training images as unrepresentative. We actually consider them as very useful tools for complex structure construction, when data (particularly vertical profiles) are available. We see that this was misleadingly formulated beforehand.

Herweijer (1997) provides a number of references to the sedimentology and paleogeography of the MADE site, which would provide a very good start with respect to representative training images. Ronayne et al. (2008) give a good example of a model based on training images to model a hydrogeological test site.

Again, we do not aim to construct a more realistic conceptual model on the costs of including observation data which is often not available at typical hydrogeological field sites.

The paper also states that geological models as used in the petroleum industry have not found their way into applied hydrogeology (line 64), a statement which is quite strong, and in my view incorrect. Alloisio, (2011), Dowling et al. (2013) and Peereboom (2018) are examples of applications for a variety of shallow to deep aquifers. Even if it is the case that this type of modelling has not 'found its way' into widespread use in hydrogeology, this is not a reason to simply to set it aside from a research point of view. The authors should explain why their approach is 'better' than the standard geological modelling methods used in the petroleum industry and the hydrogeological applications of these methods referenced earlier in this paragraph.

**The authors should re-assess the literature on the above matter and correct their statements about the use of geological data and models.**

We are happy when the referee could provide us references to papers in journals frequently read by hydrogeologists in science and practice. Given that all stated articles are conference proceeding or

theses, this somehow confirms our statement. Anyway, the corresponding paragraph was reformulated.

In this regard, we aim to clarify that we do not wish to set the work done in petroleum industry aside. Our approach is not 'better' than the standard geological modelling methods used in the petroleum industry but has a different purpose (as mentioned repeatedly and stated in the manuscript). It is well known that financial limitations are much different in (purely) hydrogeological studies. Thus, we focus on aquifer heterogeneity construction for a level of available field data usual at hydrogeological sites.

Some further issues in need of clarification:

Line 335 and last section of supplement: As earlier discussed, the MADE plumes are variable in 3D and should really be modelled in 3D. The sections explaining the 3D modelling effort to confirm the validity of the 2D model are confusing. It is unclear if the 2D model was simply copied in the 3 rd dimension, ie. is completely symmetric in the 3 rd dimension, or if some sort of heterogeneity in the 3 rd dimension was included. The supplement also states that the extension in the 3 rd dimensions has no impact because of the binary nature of the 'inclusions' vs. K values that change gradually. If transport being restricted to a 2D cross-section of a 3D model is a significant impact of the binary conceptualization, than the binary conceptualization should not be used, as gradual changes of K are the norm for non fractured/fissured aquifers such as the MADE aquifer It would also be informative if the authors would present a visualization (map or x-section slices) of the 3D model and the modelled plume. It should also be looked at if a small level of transversal dispersion (representing very small scale heterogeneity) would have a significant impact on the 3D version of the 2D model.

Data on the observed transport plumes (3d) at the 8 times of the MADE1 experiment are not available to us since they are generally not public. We therefore focused on the averaged longitudinal mass distribution (1D transects) which is available to us. We already added a comment on the data situation to section 3 (L. 245) in the previous revision.

We rewrote the paragraph concerning the model dimensionality in the manuscript and the section in the supporting information (see also comment above). We follow the referees advice, and specified the settings of the 3D inclusion model: specifically the y-direction was extended in a range of -15 to 15 m (with the source located at y=0). For an inclusion length of $I_y = 10m$, three blocks of different random inclusion structure is present in the transverse horizontal direction. We further added a figure of one realization of the 3D inclusion structure to the SM.

Since we follow a Monte Carlo approach with an ensemble of realizations, there is not a single simulated tracer plume, but a large variety which all look different depending on the binary inclusion structure realization. Our results are based on the ensemble average. To clarify that point we add here some simulated plume distributions of several realizations (of the 2D model). Note that a display of the 3D plume is hardly feasible and displaying the plume of the 3D model along the xz-cross section looks similar to those of the 2D realizations.

[Figure]

[Figure]

[Figure]

[Figure]

*Mass distribution contours (cross sectional view, color levels of 5%) for two realizations R2 & R3 at two times T=126 and 503 days after injection.*

We agree with the referee's statement on "gradual changes of K" in real aquifers. This is clearly not represented by a binary structure. We can just again repeat that we seek for a simple heterogeneity conceptualization which reproduces transport pattern sufficiently well given the defined study purpose. This does not imply that the conductivity structure actually looks realistic, which is a core characteristic of upscaling. If the purpose would have been the reconstruction of the aquifer heterogeneity in 3D and the reproduction of the 3D tracer plume, than we fully agree, that the model would need to be in 3D and that the heterogeneity would need a clearly more sophisticated conceptualization as we present. For such a study the referee is refereed to Dogan et al., 2014.

When modelling in 3D, clearly transverse horizontal dispersion takes place. Given that we resolve heterogeneity, we applied a small value for transverse horizontal dispersivity representing hydrodynamic dispersion effects solely (see also Zech et al., 2019). As expected it lead to a slight spreading of solutes in transverse horizontal direction. However, this effect is averaged out by post-processing the mass distribution to longitudinal mass transects. Again, we like to emphasize that we do not aim to provide a model which is able to give a proper plume reproduction, but an approach to reproduce main transport features.

Line 330-334 and section 'details on flow and transport' section in supplement: The injection rate modelled is Qin = 1.166e-5 m3/sec. The injection rate quoted by Boggs et. al. is 10.07 m3/48.5 hr which converts to 5.57e-5 m3/sec. If any adjustment has been made to the injection rate (perhaps to adapt for the 2D model setting?) this has to be clarified. The paper states: We use a flux related injection representing natural conditions. For technical details, the reader is referred to the Supporting Information. This is unclear and there seems to be no further specific discussion on 'flux related injection' in the supporting documentation.

The injection rate quoted by Boggs was 10.07 m3/48.5hr (=5.57e-5 m3/sec.) distributed over 5 wells (p.3285). Thus, when modelling the transport along the main flow transect, the actual injection rate in the source well is a fifth of 5.57e-5 m3/sec, thus $Q_{in}$ = 1.166e-5 m3/sec.

We thought the reader familiar with typical injection modes (initial conditions) of transport models, which are typically in resident or flux proportional mode [Kreft and Zuber, 1978]: for a resident mode the initial mass is constant (e.g. along the well) while for a flux proportional mode it is distributed according to the local conductivity (e.g. along the well). When dealing with transport in heterogeneous media, it is particularly important to distinguished according to the given field conditions.

We specified both information in the SM, which were indeed missing so far.

Section 3.4: calibration and predictive capacity of model: The model matches data in a line downstream of the injection, but as discussed earlier in the review (point 2) does not represent any 3D movement of the plume close to the injection point and further downstream. The paper also does not show a vertical cross-section of the observed vs the modelled tracer distribution. Hence any calibration is quite limited and potentially has limited predictive value As discussed in the paper (line 270) there is significant uncertainty due to the mass balance issues with the bromide tracer (~50% in snapshots past 300 days). The calibration could be (should be?) tested using the MADE

2 tritium tracer test (Boggs et al., 1993), which seems to have resulted in a better mass balance (77% in final snapshot at 328 days).

We agree that our model does not provide information on mass distribution at point resolution level. Again, we did not aim to reproduce the actual plume distribution. The model matches the data in a line downstream of the injection, which is the perpendicularly averaged mass distribution along the main flow path. Thus, it should be considered as the average of observed values along a observation transect located at the distance x from the injection. When aiming to predict major plume properties such as leading mass and location of the bulk mass, this is highly useful information. In this line, we also see no point in comparing vertical cross-sections of modelled and observed data, which is anyway not possible for us since the concentration distribution data is not public available for all time snapshots of the MADE1 tracer experiment.

Again, we do NOT calibrate the model. We refrain from calibration to keep the model predictive mimicking field situation where no preliminary information on transport behaviour is available. We also refrain from repeating the study focusing on MADE-2 experimental results, which show in general the same pattern, although having a slightly better mass recovery. As the referee might be aware of, there are other issues with the MADE-2 experimental data, such as transient flow conditions due to significant seasonal water table fluctuations during the experiment. However, we are sure when running flow simulation adapted to the MADE-2 experimental settings for an ensemble of random conductivity with the binary inclusion structure, the observed longitudinal mass distribution would show the same characteristics as observed in the MADE-2 experiment, which are generally similar to those of the MADE-1 experiment.

Title: There is a grammar error in the title (the part which reads 'in a Heterogeneous Aquifers'). It should be either 'in a Heterogeneous Aquifer' or 'in Heterogeneous Aquifers'

We thank the reviewer for pointing out this flaw. We corrected it.

[revised manuscript text omitted]